# ZFYVE28 mediates insulin resistance by promoting phosphorylated insulin receptor degradation via increasing late endosomes production

Liang Yu[1,3], Mengchen Xu[1,3], Yupeng Yan[1,3], Shuchen Huang[1,3], Mengmeng Yuan[1], Bing Cui[1], Cheng Lv[1], Yu Zhang[1], Hongrui Wang[1], Xiaolei Jin[2], Rutai Hui[1] & Yibo Wang [1] ✉

Insulin resistance is associated with many pathological conditions, and an in-depth understanding of the mechanisms involved is necessary to improve insulin sensitivity. Here, we show that ZFYVE28 expression is decreased in insulin-sensitive obese individuals but increased in insulin-resistant individuals. Insulin signaling inhibits ZFYVE28 expression by inhibiting NOTCH1 via the RAS/ERK pathway, whereas ZFYVE28 expression is elevated due to impaired insulin signaling in insulin resistance. While Zfyve28 overexpression impairs insulin sensitivity and causes lipid accumulation, Zfyve28 knockout in mice can significantly improve insulin sensitivity and other indicators associated with insulin resistance. Mechanistically, ZFYVE28 colocalizes with early endosomes via the FYVE domain, which inhibits the generation of recycling endosomes but promotes the conversion of early to late endosomes, ultimately promoting phosphorylated insulin receptor degradation. This effect disappears with deletion of the FYVE domain. Overall, in this study, we reveal that ZFYVE28 is involved in insulin resistance by promoting phosphorylated insulin receptor degradation, and ZFYVE28 may be a potential therapeutic target to improve insulin sensitivity.

Obesity is one of the major public health problems globally with the prevalence of obesity tripling since the mid-1970s; 650 million adults and 124 million children and adolescents are obese, which is responsible for some cancers, most type 2 diabetes and almost 40% of cardiovascular diseases[1,2]. Insulin resistance is one of the common complications in obese people and is closely related to the development of a variety of diseases, including metabolic syndrome (MetS), dyslipidemia and hypertension[3]. However, although most obese patients show impaired insulin sensitivity, approximately 30% of obese patients are known as metabolically healthy obesity in clinical observations[4]. They still maintain a normal metabolic index with insulin sensitivity similar to healthy individuals, while the insulin levels are compensatorily increased[5–7]. The differences in biochemical and molecular regulatory mechanisms between this population and insulin-resistant obese patients are not well understood. An in-depth understanding of the mechanisms of insulin resistance is necessary and important to improve insulin sensitivity and prevent associated metabolic and cardiovascular diseases.

FYVE domain is a double zinc finger-like domain that predominantly binds phosphatidylinositol 3-phosphate (PI3P) and

[1]State Key Laboratory of Cardiovascular Disease, Fuwai Hospital, National Center for Cardiovascular Diseases, Chinese Academy of Medical Sciences and Peking Union Medical College, Beijing, China. [2]Plastic Surgery Hospital, Chinese Academy of Medical Sciences & Peking Union Medical College, Beijing, China. [3]These authors contributed equally: Liang Yu, Mengchen Xu, Yupeng Yan, Shuchen Huang. ✉e-mail: yibowang@hotmail.com

mediates endosomal transport[8–10]. The FYVE domain contains a PI3P-binding pocket and localizes and activates the binding pocket through nonspecific insertion of a hydrophobic loop into the lipid bilayer[11]. FYVE domain-containing proteins are also involved in intracellular trafficking and signal transduction[12]. ZFYVE28, zinc finger FYVE-type containing 28, is a major member of the FYVE domain-containing proteins. It has been shown that ZFYVE28 is able to regulate the endosomal localization of epidermal growth factor receptor (EGFR), a receptor tyrosine kinase, and transfer it to lysosomes for degradation[13]. Insulin receptor (INSR) is also a receptor tyrosine kinase like EGFR. However, how ZFYVE28 regulates the INSR and how it is involved in insulin signaling remains unclear.

In this study, we show that ZFYVE28 is involved in insulin signaling regulation and insulin signaling inhibits ZFYVE28 expression by inhibiting NOTCH1. We also investigate the effect of Zfyve28 on insulin sensitivity in mice, showing that Zfyve28 knockout in mice can significantly improve insulin sensitivity and other indicators associated with insulin resistance. We further confirm that ZFYVE28 colocalizes with early endosomes via the FYVE domain, which promotes the conversion of early to late endosomes and promotes the degradation of phosphorylated insulin receptor. Our study reveals a detailed mechanism by which ZFYVE28 regulates insulin signaling, and ZFYVE28 may be a potential therapeutic target to improve insulin sensitivity.

## Results

### ZFYVE28 expression was lower in obese-IS subjects but higher in insulin-resistant subjects

We first recruited an obese cohort with normal insulin sensitivity (insulin-sensitive obese subjects, hereafter referred to as obese-IS), a MetS cohort with insulin resistance and another 100 matched normal controls with strict criteria. The age ranged from 50–77, and the controls had normal BMI (18.5–23.9). The patients with obesity had BMI ≥ 28.0 or BMI > 27.0 and waist circumference > 101 cm. All obese patients were nondiabetic and had normal insulin sensitivity. The diagnostic criteria for MetS referred to the American Heart Association definition of MetS[14]. Specifically, the MetS cohort met the criteria for the obese population described above plus ≥3 of the following: (1) elevated triglycerides (drug treatment for elevated triglycerides was an alternate indicator) ≥1.7 mmol/L; (2) reduced HDL-C (drug treatment for reduced HDL-C was an alternate indicator) < 1.0 mmol/L; (3) elevated blood pressure (antihypertensive drug treatment in a patient with a history of hypertension was an alternate indicator) with current or previous systolic blood pressure (SBP) ≥ 160 mmHg and diastolic blood pressure (DBP) ≥ 100 mmHg; and (4) elevated fasting glucose (drug treatment of elevated glucose was an alternate indicator) ≥ 6.1 mmol/L. All MetS patients showed insulin resistance. The basic characteristics are shown in Supplementary Table 1. Second, 13 control males, 13 obese-IS patients and 13 insulin-resistant MetS patients were randomly selected for gene expression profiling analysis by using Affymetrix Human Gene 2.0ST Array (Fig. 1a); the basic characteristics are shown in Supplementary Table 2. Both obese-IS and MetS patients had elevated insulin levels, which were higher in MetS patients (Supplementary Table 2). The volcano plots illustrated the differentially expressed genes, and the results showed that *ZFYVE28* expression was lower in the peripheral blood of the obese-IS patients but higher in MetS patients with insulin resistance (Fig. 1b, d), which was further verified in the remaining samples (87 controls vs. 87 obese-IS patients, 87 controls vs. 87 MetS patients) by RT-qPCR (Fig. 1c, e). In addition, we obtained 10 human fat samples (3 nonobese noninsulin-resistant samples, 4 insulin-sensitive obese samples, and 3 insulin-resistant samples) from patients who underwent liposuction surgery in Plastic Surgery Hospital, Chinese Academy of Medical Sciences, and all patients provided informed consent. We examined the expression of ZFYVE28 RNA and protein levels in these fat samples and the results

showed that ZFYVE28 expression levels were significantly increased in fat samples from insulin-resistant patients (Supplementary Fig. 1a–c), which was consistent with the analysis results of blood samples from insulin-resistant MetS patients described above.

The above findings were intriguing, and to further investigate the role of ZFYVE28 in obesity and insulin resistance, we first analyzed the expression levels of Zfyve28 in different tissues of wild-type (WT) mice. The analysis results of mRNA and protein levels indicated that Zfyve28 was highly enriched in brain tissue from mice, while it was expressed to varying degrees in other tissues (Supplementary Fig. 1d–f). We then focused on insulin-dependent tissues such as fat, liver, and skeletal muscle, in which Zfyve28 expression levels were slightly higher in fat and liver than in skeletal muscle (Supplementary Fig. 1d–f). Next, we induced obesity and insulin resistance in mice by feeding a high-fat diet (HFD) for 4 and 12 weeks (hereafter referred to as the HFD-4w group and HFD-12w group, respectively). Compared with mice fed a normal diet (ND), obese mice in the HFD-4w group had an increase in body weight, which reached statistical significance (Fig. 1f–g). However, the obese mice did not develop significant steatosis in the liver (Fig. 1i, j). As expected, serum insulin levels were increased in the HFD-4w group, and the glucose tolerance test (GTT) and insulin tolerance test (ITT) showed normal glucose tolerance and insulin sensitivity (Fig. 1k–m). We then examined the expression of Zfyve28 in different tissues including heart, lung, brain, kidney, spleen, fat, liver, skeletal muscle and blood samples. The results indicated that *Zfyve28* mRNA expression was significantly decreased in the livers of obese-IS mice (Fig. 1n and Supplementary Fig. 1g). In contrast, the mice in the HFD-12w group had increased body weight, and the livers exhibited macroscopic steatosis (Fig. 1h, i). HE staining of liver tissues also showed large and dense fat vacuoles (Fig. 1j). Mice in the HFD-12w group had substantially elevated serum insulin levels (Fig. 1o). GTT and ITT showed impaired glucose tolerance and impaired insulin sensitivity (Fig. 1p, q). These results indicated that mice in the HFD-12w group developed insulin resistance. We subsequently examined the expression levels of Zfyve28 in different tissues, and surprisingly, *Zfyve28* mRNA expression was significantly increased in the livers of insulin-resistant mice (Fig. 1r and Supplementary Fig. 1h). We then extracted liver proteins from obese-IS and insulin-resistant mice for western blotting, further confirming that Zfyve28 was downregulated in the liver of obese-IS mice but upregulated in the liver of insulin-resistant mice (Fig. 1s–v and Supplementary Fig. 1i–j).

Taken together, these findings suggested that ZFYVE28 expression was lower in obese subjects with normal insulin sensitivity, while it was elevated in subjects with insulin resistance.

### ZFYVE28 expression was inhibited by insulin via the RAS/ERK pathway in HepG2 cells and primary hepatocytes

This finding of differential expression of ZFYVE28 in the two conditions of obesity and insulin resistance was attractive. Taking into account differences in insulin signaling, we sought to clarify the underlying mechanism by which ZFYVE28 is regulated by insulin. Surprisingly, ZFYVE28 expression showed a dose-dependent decrease in HepG2 cells upon the treatment with insulin, which was also validated in HEK293 cells and HeLa cells (Supplementary Fig. 2a–d). This was further confirmed in mice. We analyzed liver tissues from WT mice treated with insulin (0.5 U each time) or saline for 7 consecutive days, and the results showed that Zfyve28 was inhibited under treatment with continuous high concentrations of insulin (Fig. 2a–c).

Insulin signaling includes two main pathways, the PI3K/AKT pathway and the RAS/ERK pathway, and different messenger molecules such as AKT and RAS can produce different downstream effects[15–17]. To investigate whether the inhibitory effect of insulin on ZFYVE28 occurs via the AKT pathway or RAS pathway, we treated HepG2 cells with the AKT inhibitor GSK690693 and the RAS inhibitor Pan-RAS-IN-1. ZFYVE28 expression showed no change when the AKT

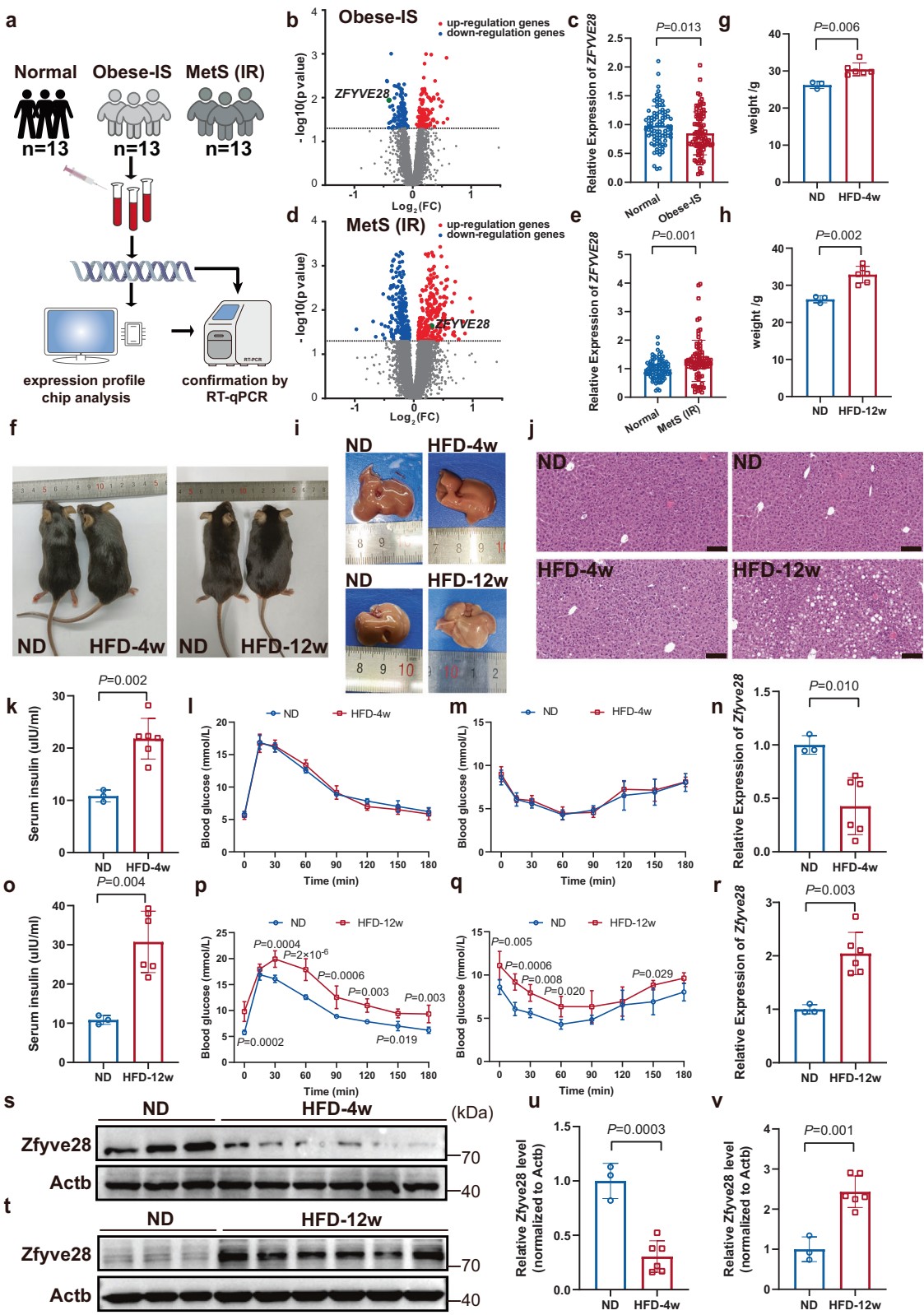

pathway was blocked, while it was significantly increased when the RAS pathway was blocked (Supplementary Fig. 2e, f). To further investigate the effect of activation and inhibition of the RAS pathway on ZFYVE28 expression, we treated HepG2 cells with the RAS inhibitor Pan-RAS-IN-1 and the RAS agonist KRA-533. We tested the active RAS level (RAS-GTP) using the GST-Raf1-RBD fusion protein. Pan-RAS-IN-1 treatment suppressed RAS activity, whereas KRA-533 treatment promoted RAS

activation (Supplementary Fig. 2g, h). Western blotting results showed that ZFYVE28 expression was significantly increased in response to Pan-RAS-IN-1 treatment, accompanied by inhibition of phospho-ERK levels (Supplementary Fig. 2i, k). The opposite results were observed with KRA-533 treatment. Activation of the RAS pathway markedly suppressed ZFYVE28 expression (Supplementary Fig. 2l–n). We further investigated the effect of the MEK inhibitor PD98059 on ZFYVE28

**Fig. 1 | ZFYVE28 expression was lower in obese-IS subjects but higher in insulin-resistant subjects. a** Test procedure for insulin-sensitive obese (obese-IS) patients, insulin-resistant MetS (IR) patients and normal controls ($n = 13$ per group). MetS, metabolic syndrome. **b**, **d** Volcano plots illustrated the differentially expressed genes in whole blood cells between controls and obese-IS patients (**b**), as well as controls and MetS patients (**d**). Compared to controls, *ZFYVE28* expression was decreased in obese patients (FC = 0.75, $P = 0.011$) but increased in insulin-resistant patients (FC = 1.24, $P = 0.023$). Points corresponding to significantly downregulation and upregulation genes are colored blue and red, respectively. *ZFYVE28* is highlighted with a green dot. Fold change is displayed in a log2 scale and $P$ values in −log10 ($P$ value); FC fold change. **c**, **e** The expression difference of *ZFYVE28* in the remaining samples was further verified by RT-qPCR (87 controls vs. 87 obese-IS patients, 87 controls vs. 87 MetS patients). **f**–**j** Gross images (**f**), body weight (**g**, **h**), liver images (**i**) and HE staining results of liver tissues (**j**) of mice in the HFD-4w group and HFD-12w group. ND: $n = 3$; HFD-4w: $n = 6$; HFD-12w: $n = 6$ biologically independent mice. Scale bar, 100 μm. **k**, **o** Serum insulin levels of mice in the HFD-4w group (**k**) and HFD-12w group (**o**). ND: $n = 3$; HFD-4w: $n = 6$; HFD-12w: $n = 6$ biologically independent mice. **l**, **m**, **p**, **q** GTT and ITT results showed normal glucose tolerance (**l**) and insulin sensitivity (**m**) in HFD-4w group mice, as well as impaired glucose tolerance (**p**) and insulin sensitivity (**q**) in HFD-12w group mice. ND: $n = 3$; HFD-4w: $n = 6$; HFD-12w: $n = 6$ biologically independent mice. **n**, **r** *Zfyve28* expression was downregulated in the liver of obese-IS mice in the HFD-4w group (**n**) but upregulated in the liver of insulin-resistant mice in the HFD-12w group (**r**). ND: $n = 3$; HFD-4w: $n = 6$; HFD-12w: $n = 6$ biologically independent samples. **s**–**v** Western blot assays of Zfyve28 expression in the livers of mice in the HFD-4w group (**s**) and HFD-12w group (**t**), and the quantitative analysis results (**u**, **v**) are presented. ND: $n = 3$; HFD-4w: $n = 6$; HFD-12w: $n = 6$ biologically independent samples. Data are shown as means ± SD, and $P$ values are determined by unpaired two-tailed Student's $t$-test (**b**–**e**, **g**, **h**, **k**, **n**–**o**, **r**, **u**, **v**) or two-way ANOVA with Fisher's LSD post hoc multiple comparisons test (**l**, **m**, **p**, **q**). The exact $P$ values are shown in the figure. Source data are provided as a Source Data file.

expression levels, which showed similar effects as the RAS inhibitor. PD98059 treatment in HepG2 cells inhibited the phosphorylation levels of ERK in a dose-dependent manner, accompanied by elevated ZFYVE28 expression levels (Supplementary Fig. 2o–q).

To investigate the expression differences of ZFYVE28 in insulin resistance, we utilized palmitic acid (PA)-treated HepG2 cells to imitate impaired insulin sensitivity[18–20]. Cells treated with PA had less glycogen content and showed little response to insulin stimulation with no change in glycogen content, while insulin stimulation greatly increased the glycogen content in normal cells (Supplementary Fig. 3a, b). Western blotting results further confirmed impaired insulin signaling, as demonstrated by decreased phospho-insulin receptor (p-INSR) levels, with decreased phospho-AKT (p-AKT) and phospho-ERK (p-ERK) levels (Supplementary Fig. 3c, f–h). This suggested that PA treatment impaired normal insulin signaling within cells. Moreover, ZFYVE28 expression was upregulated with PA treatment (Supplementary Fig. 3c, e). We then treated HepG2 cells with insulin, KRA-533 and PA. The results of the RAS activity assay and western blotting of phosphorylated ERK showed that both insulin and KRA-533 treatment significantly promoted the activation of the RAS pathway, while PA treatment significantly inhibited the RAS pathway (Fig. 2d, f and Supplementary Fig. 3m,). With PA-induced insulin resistance in cells, insulin treatment no longer caused RAS activation, while KRA-533 maintained its effect in promoting RAS activation (Fig. 2d, f and Supplementary Fig. 3m). Furthermore, intracellular glycogen content assays showed a decrease in glucose uptake with PA treatment in HepG2 cells, which was markedly increased with KRA-533 treatment (Fig. 2g, h). Western blotting results showed that both insulin and KRA-533 could significantly inhibit the expression of ZFYVE28; however, under PA-induced insulin resistance, only KRA-533 had an inhibitory effect on ZFYVE28 expression while the inhibitory effect of insulin disappeared (Fig. 2d, e). These results suggested that insulin inhibited ZFYVE28 expression via the RAS/ERK pathway, yet the inhibitory effect was abolished upon insulin resistance.

We further confirmed these results in primary hepatocytes. We first examined hepatic protein levels of obese-IS and insulin-resistant mice. Western blotting showed increased levels of phosphorylated insulin receptor (p-Insr) in the livers of obese-IS mice in the HFD-4w group, in which Zfyve28 expression was decreased (Figs. 1n, s and 2i, k). The levels of Ras-GTP, the active form of Ras, and phosphorylated Erk also increased (Fig. 2i, l, m). However, as shown by decreased phosphorylated Insr and phosphorylated Erk levels, insulin signaling was markedly suppressed in livers from insulin-resistant mice, in which Zfyve28 expression was increased (Figs. 1r, t and 2j, n–p). We subsequently isolated primary hepatocytes from wild-type (WT) mice and insulin-resistant mice and left the cells stably attached for experiments. Microscopically, multinucleated primary hepatocytes were observed under a microscope, and oil red O staining showed widespread lipid droplets in hepatocytes (Supplementary Fig. 4a, b). We then treated primary hepatocytes with insulin, KRA-533, Pan-RAS-IN-1 and PD98059 and examined the cellular protein levels. The results showed that insulin and KRA-533 could significantly promote the activation of the Erk pathway in primary hepatocytes of WT mice, accompanied by the inhibition of Zfyve28 expression, while the inhibition of the Erk pathway by Pan-RAS-IN-1 and PD98059 significantly increased Zfyve28 expression (Fig. 2q–s). The results were different and interesting in primary hepatocytes from insulin-resistant mice. The activation effect of insulin on the Erk pathway was almost absent, while KRA-533 treatment still maintained a good activation effect on the Erk pathway. Correspondingly, insulin no longer inhibited Zfyve28 expression, whereas KRA-533 retained its inhibitory effect on Zfyve28 (Fig. 2t–v).

In summary, these findings suggested that ZFYVE28 expression was inhibited by insulin via the RAS/ERK pathway both in vivo and in vitro, and the inhibitory effect disappeared when insulin sensitivity was impaired, which could be re-inhibited by the RAS agonist KRA-533.

## ZFYVE28 was transcriptionally regulated by NOTCH via RBP-Jk

The activation of RAS has been shown to downregulate the NOTCH pathway[21,22], and the activation of the NOTCH pathway has also been shown to be strongly associated with insulin resistance[23–25]. We extracted liver RNA from obese-IS mice and insulin-resistant mice and then performed transcriptome sequencing. Gene set enrichment analysis (GSEA) showed that the Notch pathway was downregulated in obese-IS mouse livers in the HFD-4w group and upregulated in insulin-resistant mouse livers in the HFD-12w group (Fig. 3a–d). Western blotting results also showed that Notch1 levels were decreased in the livers of obese-IS mice, but increased in the livers of insulin-resistant mice (Fig. 3e–h). Thus, it could be speculated whether ZFYVE28 was transcriptionally regulated by NOTCH. We also found that in HepG2 cells treated with insulin, HES1 and HEY1, the downstream genes of NOTCH, as well as ZFYVE28 decreased at the mRNA level (Fig. 3i). To determine whether the downregulation of ZFYVE28 was caused by the inhibition of transcription initiation or the degradation of mRNA, we utilized actinomycin D (ActD) to block transcription initiation and found that insulin did not alter the half-life of ZFYVE28 mRNA (Fig. 3j). These results suggested that insulin could inhibit the transcription initiation of ZFYVE28 by downregulating NOTCH signaling.

In the classical NOTCH signaling pathway, the NOTCH protein is cleaved three times when activated and then the intracellular segment (NICD) is released into the cytoplasm. After binding to the transcription factor CSL, such as RBP-Jk, an NICD/CSL transcriptional activation complex is formed and promotes the expression of downstream genes[26]. We examined the effect of insulin on NICD, and the intracellular NICD was decreased with insulin stimulation (Fig. 3k). We further

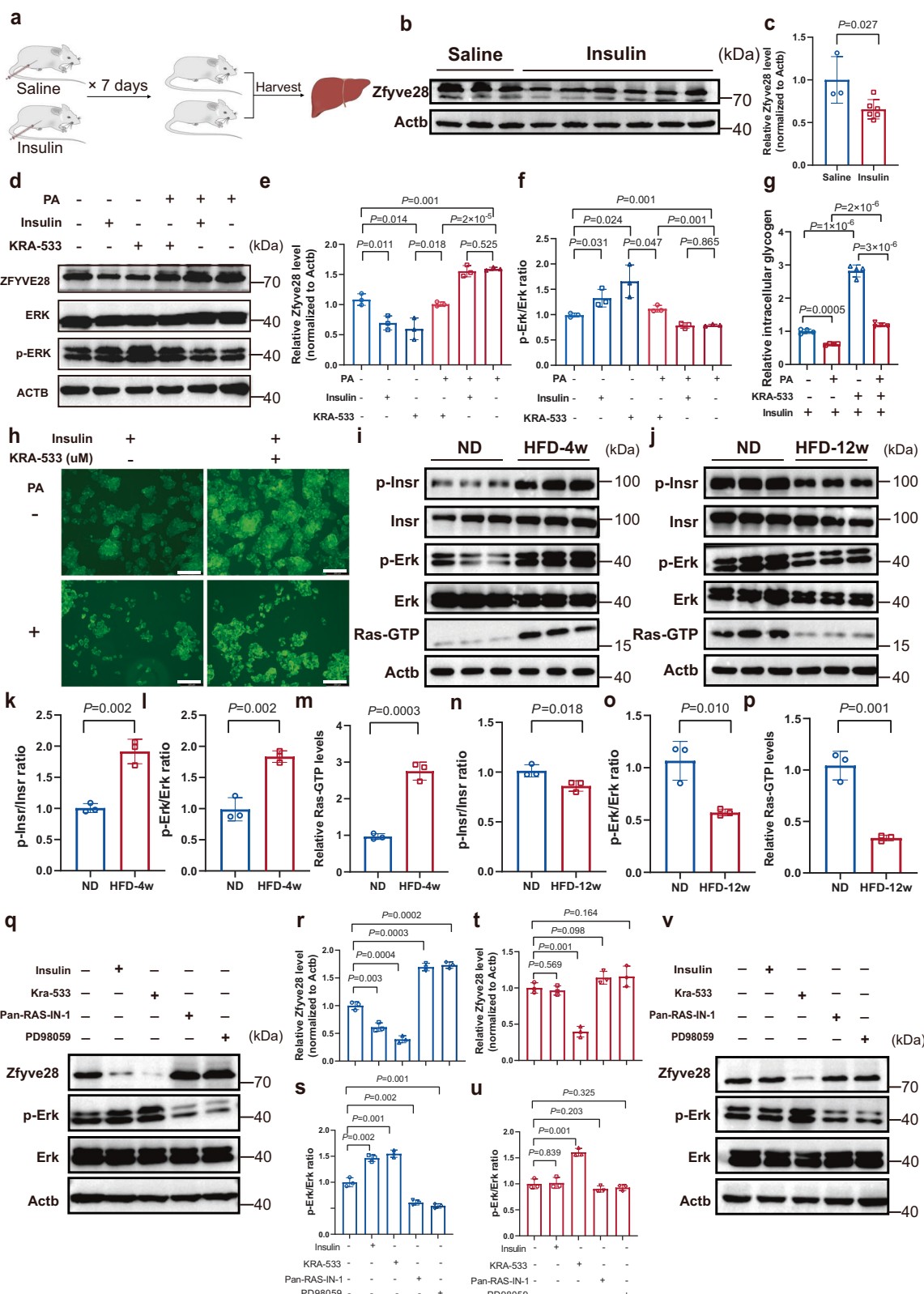

validated the effect of NOTCH signaling on ZFYVE28 by constructing a cell model overexpressing NOTCH1, which resulted in a significant increase in ZFYVE28, HES1 and HEY1 expression (Fig. 3l). As expected, knockdown of NOTCH1 in HepG2 cells caused a decrease in the expression levels of ZFYVE28, as well as HES1 and HEY1 (Fig. 3m). To determine whether the transcription initiation of ZFYVE28 was directly regulated by NOTCH, we used JASPAR to predict possible transcription

factors of ZFYVE28 and found a possible binding site of RBP-Jk (Fig. 3n). Then, we confirmed that RBP-Jk could indeed bind to the ZFYVE28 promoter by chromatin coimmunoprecipitation (ChIP) (Fig. 3o). We also constructed a dual luciferase reporter system by inserting the *ZFYVE28* promoter before the firefly luciferase reporter gene, and the results showed that NOTCH1 overexpression clearly enhanced the luciferase activity (Fig. 3p, q). We also examined NOTCH1

**Fig. 2 | ZFYVE28 expression was inhibited by insulin via the RAS/ERK pathway in HepG2 cells and primary hepatocytes. a–c** Schematic diagram of mice treated with saline or insulin (**a**) and western blot results of Zfyve28 in the liver (**b**); insulin bioactivity was 0.5 U for each injection. The quantitative analysis results (**c**) are shown. Saline: $n = 3$; Insulin: $n = 6$ biologically independent samples. **d–f** Representative western blot results (**d**) of HepG2 cells treated with PA, insulin, and KRA-533; the quantitative analysis results of three independent parallel experiments (**e, f**) are shown. PA, palmitic acid. **g, h** Intracellular glycogen content assay results of HepG2 cells treated with/without PA and/or KRA-533. Representative images (**h**) and quantitative analysis results (**g**) are presented; $n = 4$ biologically independent samples per group. Scale bar, 200 μm. **i–p** Western blot analysis of hepatic protein levels in the HFD-4w group mice (**i**) and HFD-12w group mice (**j**).

The quantitative analysis results (**k–p**) are shown; $n = 3$ biologically independent samples per group. **q–s** Primary hepatocytes were isolated from wild-type mice. Representative western blotting assay results of primary hepatocytes treated with insulin, KRA-533, Pan-RAS-IN-1 and PD98059 (**q**) are shown, as well as quantitative analysis results of three independent parallel experiments (**r, s**). **t–v** Primary hepatocytes were isolated from insulin-resistant mice. Representative western blotting assay results of primary hepatocytes treated with insulin, KRA-533, Pan-RAS-IN-1 and PD98059 (**v**) are shown, as well as quantitative analysis results of three independent parallel experiments (**t, u**). Data are shown as means ± SD, and *P* values are determined by unpaired two-tailed Student's *t*-test (**c, e–g, k–p, r–u**). The exact *P* values are shown in the figure. Source data are provided as a Source Data file.

expression in the cell models described above. HepG2 cells treated with the RAS inhibitor showed increased NOTCH1 expression while RAS agonist treatment resulted in decreased NOTCH1 expression (Supplementary Fig. 3i–l). As expected, NOTCH1 expression was increased and the inhibitory effect of insulin was attenuated in HepG2 cells treated with PA, while the RAS agonist could still inhibit NOTCH1 expression (Supplementary Fig. 3n–o). The levels of NICD, the active form of NOTCH1, also showed similar changes (Fig. 3r, s). Similar to RAS inhibitor treatment, MEK inhibitor treatment also caused a significant increase in NICD levels (Supplementary Fig. 3p, q).

We then performed validation in mouse primary hepatocytes. Consistent with the expression pattern of Zfyve28, treatment with insulin and KRA-533 markedly decreased Nicd levels in primary hepatocytes from WT mice, which rose following Pan-RAS-IN-1 and PD98059 treatment (Fig. 3t, u). However, only KRA-533 inhibited Nicd in hepatocytes from insulin-resistant mice (Fig. 3v, w).

In summary, these results indicated that NOTCH regulated the transcription initiation of ZFYVE28, which was inhibited by insulin.

## Global *Zfyve28* knockout improved insulin sensitivity in mice

The above results implied a link between ZFYVE28 expression and insulin signaling. To confirm this, we generated *Zfyve28* global knockout mice (Fig. 4a). Intriguingly, *Zfyve28* heterozygous (*Zfyve28*[+/−]) mice and *Zfyve28* global knockout mice exhibited a gene dose-response relationship of Zfyve28 on insulin sensitivity. In *Zfyve28*[+/−] mice, Zfyve28 was mostly knocked down (Supplementary Fig. 5a, b). When fed a HFD, *Zfyve28*[+/−] mice showed a decreased trend in body weight compared to WT mice, but this difference was not significant (Supplementary Fig. 5c). GTT and ITT showed that *Zfyve28*[+/−] mice exhibited slightly improved glucose tolerance and insulin sensitivity (Supplementary Fig. 5d, e). However, livers from *Zfyve28*[+/−] mice also showed a steatotic appearance, with HE staining results showing numerous and dense fat vacuoles (Supplementary Fig. 5f, g). Western blotting results of liver proteins showed that heterozygous *Zfyve28* knockdown partially enhanced insulin signaling (Supplementary Fig. 5h–l).

In contrast, both mRNA and protein levels showed nearly complete knockout of Zfyve28 in *Zfyve28* global knockout (KO) mice (Fig. 4b, c). We then injected insulin (2 U/kg body weight) intraperitoneally into WT and KO mice and collected liver tissues for analysis 30, 60 and 90 min after injection. The results showed that the levels of phosphorylated Insr and phosphorylated Erk were higher in KO mice than WT mice at 90 min (Fig. 4d–f). This suggested that *Zfyve28* knockout somewhat maintained longer insulin signaling and delayed the degradation of phosphorylated Insr. Moreover, when fed a HFD, KO mice showed a significant decrease in body weight compared to WT mice (Fig. 4g). GTT and ITT also showed a significant improvement in glucose tolerance and insulin sensitivity in KO mice (Fig. 4h, i). Excitingly, *Zfyve28* knockout was also somewhat prevented hepatosteatosis under HFD induction, and HE staining of tissue sections showed fewer fat vacuoles (Fig. 4j, k). Western blotting analysis of hepatic protein levels showed that KO mice had significantly elevated

phosphorylated Insr (Fig. 4l, m). The downstream signaling of phosphorylated Insr was also significantly enhanced, as shown by higher levels of phosphorylated Irs1, phosphorylated Akt and phosphorylated Erk, implying that insulin signaling was markedly enhanced in KO mice (Fig. 4l, n–p).

Taken together, there was a gene dose-response relationship of Zfyve28 on insulin sensitivity and *Zfyve28* KO mice demonstrated significant improvement in insulin sensitivity.

## Liver-specific *Zfyve28* overexpression impaired insulin sensitivity and led to worse indicators associated with insulin resistance in mice

The ameliorating effect of *Zfyve28* global knockout on insulin signaling demonstrated the intervention effect of targeting Zfyve28 on insulin sensitivity in vivo. We previously elucidated that Zfyve28 expression was higher in the livers of insulin-resistant mice (Fig. 1r, t). To further investigate the effect of ZFYVE28 on insulin sensitivity, we generated a mouse model with liver-specific overexpression of *Zfyve28* (replaced by *Zfyve28*-LOE in the subsequent narrative) by tail vein injection of AAV9-pTBG-Zfyve28, in which liver-specific infection was achieved by a combination of the serotype AAV9, a serotype that efficiently infected the liver of mice, and liver-specific promoter pTBG (Supplementary Fig. 6a). The detection results of RNA and protein levels confirmed the specific overexpression of Zfyve28 in liver tissues (Supplementary Fig. 6b, d). After HFD feeding, control mice and *Zfyve28*-LOE mice showed no differences in food intake, nor did they differ much in body weight (Supplementary Fig. 6c, e). However, GTT and ITT results showed that *Zfyve28*-LOE markedly impaired glucose tolerance and insulin sensitivity in mice (Supplementary Fig. 6f–g). HE staining of liver tissues from *Zfyve28*-LOE mice showed larger and more fat vacuoles (Supplementary Fig. 6h). Western blotting results also showed that *Zfyve28*-LOE significantly inhibited insulin signaling (Supplementary Fig. 6i–m).

We then examined the effect of Zfyve28 liver-specific overexpression on other metabolic and cardiovascular indicators associated with insulin resistance in mice. We found that *Zfyve28*-LOE caused an increase tendency in liver weight and a higher liver/body weight ratio (Supplementary Fig. 6n–o). In *Zfyve28*-LOE mice, heart weight and heart/body weight ratio did not show any changes, but there was a tendency to increase (Supplementary Fig. 6p, q). In addition, the serum levels of triglycerides (TG) and total cholesterol (TC) were significantly increased in *Zfyve28*-LOE mice (Supplementary Fig. 6r, s). We also examined the lipid content in the liver and heart and found that *Zfyve28*-LOE significantly promoted triglyceride accumulation in the liver and heart (Supplementary Fig. 6t, v), while it also caused an increase in total cholesterol content in the liver, but had little effect on cardiac cholesterol content (Supplementary Fig. 6u, w). Next, the microcirculatory blood flow in the heart was monitored using laser Doppler flowmetry, and we found that *Zfyve28*-LOE caused a decrease in microcirculatory blood flow in the mouse heart, which appeared to be associated with increased levels of triglycerides in the heart (Supplementary Fig. 6x, y). Finally, the blood pressure of mice

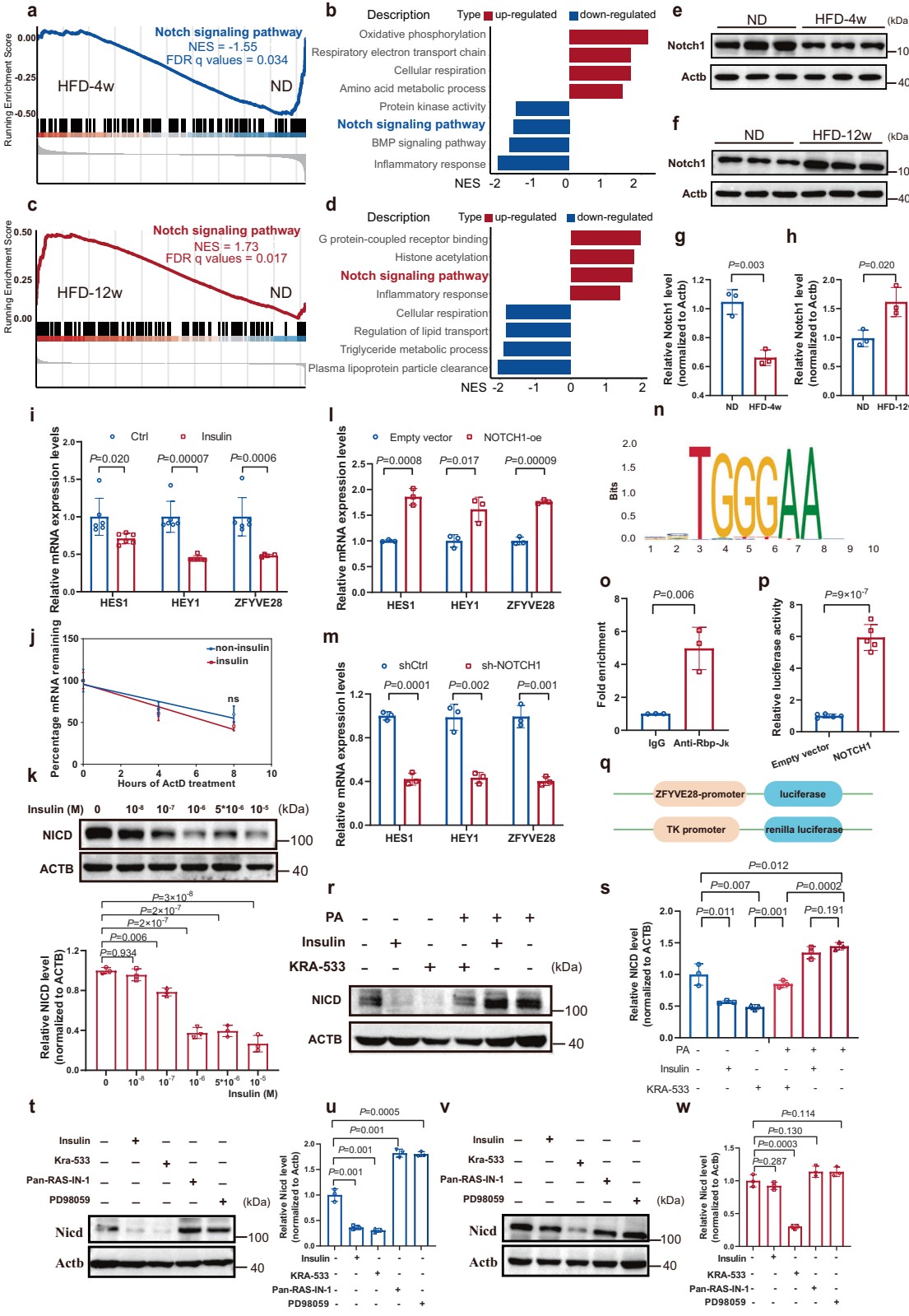

was examined using the IITC MRBP system. The results showed that DBP was significantly increased in *Zfyve28*-LOE mice compared to controls, whereas SBP showed an upward trend (Supplementary Fig. 6z).

Overall, the liver-specific overexpression of *Zfyve28* markedly impaired insulin sensitivity and led to worse metabolic and cardiovascular indicators associated with insulin resistance in mice.

## The liver-specific knockout of *Zfyve28* markedly improved insulin sensitivity and other indicators associated with insulin resistance in mice

The above results were surprising and further suggested the possibility of intervention with Zfyve28 on improving insulin sensitivity. We next generated *Zfyve28* liver-specific knockout mice (hereafter denoted as LKO mice) (Fig. 5a). The expression of Zfyve28 was detected in tissues

**Fig. 3 | ZFYVE28 was transcriptionally regulated by NOTCH via RBP-Jκ.**
**a–d** GSEA of the liver transcriptome sequencing results showed that the Notch pathway was suppressed in insulin-sensitive obese mice in the HFD-4w group (**a**, **b**) but activated in insulin-resistant mice in the HFD-12w group (**c**, **d**); $n = 3$ biologically independent samples per group. The $P$ values in GSEA are calculated empirically with permutation test by permuting the gene labels at random according to the GSEA algorithm, and FDR $q$ values (corrected $P$ values) are calculated by Benjamini–Hochberg FDR correction. The NES and FDR $q$ values are shown in the images. NES normalized enrichment score, FDR false positive rate, GSEA gene set enrichment analysis. **e–h** Western blotting analysis of Notch1 protein in livers from HFD-4w group mice (**e**) and HFD-12w group mice (**f**). The quantitative analysis results (**g**, **h**) are shown; $n = 3$ biologically independent samples per group. **i** Relative mRNA expression of *HES1*, *HEY1* and *ZFYVE28* in HepG2 cells treated with insulin at $10^{-6}$ M; $n = 6$ biologically independent samples per group. **j** Percentage mRNA remaining of *ZFYVE28* in HepG2 cells treated with ActD in the presence or absence of insulin stimulation at $10^{-6}$ M; $n = 3$ biologically independent samples per group. **k** Representative western blot results of NICD in HepG2 cells treated with insulin; the quantitative analysis results of three independent parallel experiments

are shown. **l**, **m** The expression levels of *HES1*, *HEY1* and *ZFYVE28* following NOTCH1 overexpression (**l**) and NOTCH1 knockdown (**m**) in HepG2 cells; $n = 3$ biologically independent samples per group. **n** RBP-Jk binding site prediction in the ZFYVE28 promoter region using JASPAR. **o** Chromosome immunoprecipitation assay showed the fold enrichment of RBP-Jk in the *ZFYVE28* promoter region; $n = 3$ biologically independent samples per group. **p**, **q** Diagram of the dual luciferase reporter system (**q**) and results of the luciferase activity assays (**p**); $n = 5$ biologically independent samples per group. **r**, **s** Representative western blotting results of NICD levels (**r**) in HepG2 cells treated with insulin and KRA-533. The quantitative analysis results of three independent parallel experiments (**s**) are shown. **t–w** Representative western blotting results of Nicd levels, as well as quantitative analysis results of three independent parallel experiments, in primary hepatocytes from WT mice (**t**, **u**) and insulin-resistant mice (**v**, **w**). Data are shown as means ± SD, and $P$ values are determined by unpaired two-tailed Student's $t$-test (**g–j**, **l**, **m**, **o**, **p**, **s**, **u**, **w**) or one-way ANOVA followed by Tukey's post hoc tests (**k**). The exact $P$ values are shown in the figure. Source data are provided as a Source Data file.

including heart, lung, liver, fat and skeletal muscle, and the results showed that hepatic Zfyve28 was specifically knocked out in LKO mice (Fig. 5b, d). When fed a HFD, there was no difference in food intake between control and LKO mice, but LKO mice weighed significantly less than controls (Fig. 5c, e). GTT and ITT results indicated better glucose tolerance and insulin sensitivity in LKO mice (Fig. 5f, g). In addition, livers from LKO mice did not exhibit visually visible steatosis similar to the livers of HFD-fed control mice, and HE staining also showed reduced fat vacuoles (Fig. 5h). Western blotting analysis of liver proteins showed enhanced insulin signaling in the livers of LKO mice, as shown by increased phosphorylation levels of Insr and downstream signaling molecules (Fig. 5i–m). The test results of relevant metabolic and cardiovascular indicators associated with insulin resistance were even more promising. Liver weight, heart weight, liver/body weight ratio, and heart/body weight ratio were significantly lower, suggesting that organ burden induced by HFD was improved in LKO mice (Fig. 5n–q). In addition, triglyceride and total cholesterol levels in the serum, liver and heart were also decreased in LKO mice (Fig. 5r–w). An increase in microcirculatory blood flow in the hearts of LKO mice was further confirmed by laser Doppler blood flow measurements (Fig. 5, y). Liver-specific *Zfyve28* knockout also significantly reduced SBP and DBP in mice (Fig. 5z). In summary, liver-specific knockout of *Zfyve28* in mice significantly improved insulin sensitivity and other relevant indicators associated with insulin resistance.

### ZFYVE28 promoted phosphorylated INSR degradation by targeting early endosomes via the FYVE domain

Compared to the ND group, phosphorylated Insr levels were higher in obese-IS mice with lower expression of Zfyve28, while they were lower in insulin-resistant mice with higher expression of Zfyve28 (Figs. 1s, t and 2i, j). Lower phosphorylated Insr levels were also observed in *Zfyve28* LOE mice, which were higher in *Zfyve28* LKO mice (Fig. 5i and Supplementary Fig. 6i). Therefore, we speculated that ZFYVE28 might have a role in promoting the degradation of phosphorylated INSR in our study.

To test this hypothesis, we stably overexpressed ZFYVE28 in HepG2 cells by lentiviral transfection (Fig. 6a). After insulin stimulation of HepG2 cells, ERK and its phosphorylation levels were measured at different time points. Cells overexpressing ZFYVE28 tended to have a decrease in phospho-ERK levels at the same time points (Supplementary Fig. 7a, b). We further examined INSR and phosphorylated INSR levels at different time points. Cells overexpressing ZFYVE28 exhibited a clear and consistent decrease in phosphorylated INSR; in other words, cells overexpressing ZFYVE28 had faster degradation of phosphorylated INSR than controls (Fig. 6b, c). We also examined the localization of endosomes in cells, and after lentivirus infection,

fluorescence staining of HepG2 cells revealed that ZFYVE28 (carrying flag tags) colocalized with early endosomes but not the empty vector (Supplementary Fig. 8a). We used insulin endocytosed into cells to represent internalized phosphorylated INSR, and further fluorescence staining revealed that ZFYVE28 overexpression increased insulin colocalization with early endosomes compared with controls (Supplementary Fig. 8b). Then we knocked down ZFYVE28 in HepG2 cells, and the decrease in ZFYVE28 delayed the degradation of phosphorylated INSR, which showed higher levels of phosphorylated INSR at the same time points after insulin stimulation compared with controls (Fig. 6d–f).

The FYVE domain contains a positively charged pocket that binds PI3P and inserts into the lipid bilayer via its hydrophobic loop to mediate endosomal localization[11]. Based on previous studies, ZFYVE28 mediates the endosomal localization of EGFR via the FYVE domain and then promotes the degradation of EGFR[13,27]. The FYVE domain seemed to play an important role in this process. We further constructed a ZFYVE28 mutant with deletion of the FYVE domain (ZFYVE28-ΔFYVE) (Fig. 6g). Compared to controls, overexpression of ZFYVE28-ΔFYVE did not alter the rate of degradation of phosphorylated INSR (Fig. 6h, i).

To further verify whether ZFYVE28 promoted phosphorylated INSR degradation or dephosphorylation, we treated cells overexpressing ZFYVE28 with cycloheximide (CHX) to block intracellular protein synthesis and examined total INSR levels at different time points. Western blotting results showed that INSR levels gradually decreased with increasing CHX treatment time and dropped to less than half after 6 h of CHX treatment, which implied that ZFYVE28 might promote phosphorylated INSR degradation rather than dephosphorylation (Fig. 6j, k). We then subjected ZFYVE28-overexpressing cells and control cells to prolonged CHX treatment and assayed and compared the levels of total INSR at different time points. The results showed that ZFYVE28 overexpression clearly promoted INSR degradation (Fig. 6l, m). Interestingly, ZFYVE28 overexpression maintained a facilitative effect on receptor degradation compared to ZFYVE28-ΔFYVE, suggesting a key role for the FYVE structural domain in this process (Supplementary Fig. 7c, d). With the deletion of the FYVE domain, ZFYVE28-ΔFYVE was distributed dispersively in the cell and the colocalization with early endosomes disappeared, whereas ZFYVE28 showed a concentrated intracellular distribution and tight colocalization with early endosomes (Fig. 7a). Additionally, compared to the extensive colocalization of insulin with early endosomes in ZFYVE28-overexpressing cells, there was only limited colocalization of insulin with early endosomes with deletion of the FYVE domain (Supplementary Fig. 8c). We also examined the localization of insulin with ZFYVE28. ZFYVE28-ΔFYVE was scattered

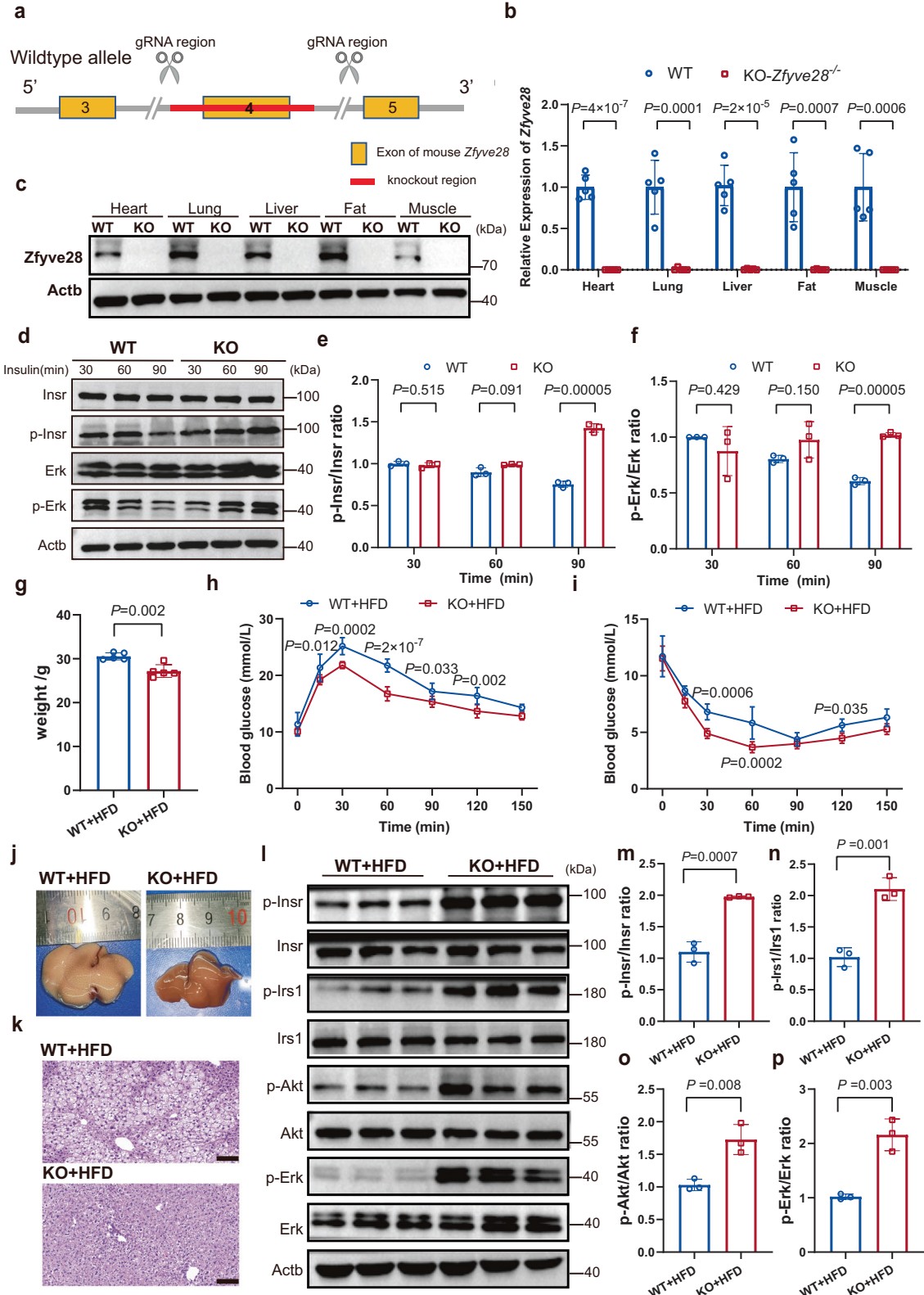

intracellularly and did not overlap significantly with insulin localization, while ZFYVE28 showed an aggregated distribution and colocalized with intracellular insulin (Supplementary Fig. 8d).

To further evaluate the promoting effect of ZFYVE28 on phosphorylated INSR degradation, we treated ZFYVE28-overexpressing and ZFYVE28-knockdown HepG2 cells with both insulin and CHX stimulation. The phosphorylated INSR and total INSR levels were then examined at various time points. The results were interesting. With prolonged CHX treatment, overexpression of ZFYVE28 significantly promoted the degradation of total INSR (Fig. 6n, p), while knockdown of ZFYVE28 significantly delayed the degradation of total INSR (Supplementary Fig. 7e, g). Furthermore, as expected, overexpression of ZFYVE28 caused a decrease in phosphorylated INSR levels in response to insulin, as indicated by lower phosphorylated INSR levels at the

**Fig. 4 | Global *Zfyve28* knockout improved the insulin sensitivity in mice.**
**a** Construction strategy of *Zfyve28* global knockout mice. **b**, **c** Zfyve28 was
knocked out in KO mice; *n* = 5 biologically independent samples per group (**b**).
Representative western blot results of three independent parallel experiments
are shown (**c**). **d**–**f** Representative western blot results (**d**) of hepatic proteins in
WT and KO mice after insulin injection (2 U/kg body weight) for 30, 60, and
90 min. The quantitative analysis results of three independent parallel experi-
ments (e-f) are shown. **g** The body weight of KO mice was lower than that of WT
mice when fed a HFD; n = 5 biologically independent mice per group. **h**, **i** GTT (**h**)
and ITT (**i**) results of WT mice and KO mice; *n* = 5 biologically independent mice

per group. **j**, **k** Global *Zfyve28* knockout prevented HFD-induced hepatosteatosis.
Representative gross images (**j**) and HE staining results (**k**) of livers are presented;
*n* = 5 biologically independent samples per group. Scale bar, 100 μm. **l**–**p** Western
blot analysis (**l**) of hepatic protein levels in WT and KO mice. The quantification
results (**m**–**p**) are shown; *n* = 3 biologically independent samples per group. Data
are shown as means ± SD, and *P* values are determined by unpaired two-tailed
Student's *t*-test (**b**, **e**–**g**, **m**–**p**) or two-way ANOVA with Fisher's LSD post hoc
multiple comparisons test (**h**, **i**). The exact *P* values are shown in the figure.
Source data are provided as a Source Data file.

same time points compared to controls (Fig. 6n, o). In contrast, when
ZFYVE28 was knocked down, phosphorylated INSR levels at the same
time point were significantly higher than in controls (Supplementary
Fig. 7e, f).

Summarizing all the above results, ZFYVE28 promoted degrada-
tion rather than dephosphorylation of the phosphorylated insulin
receptor, and the FYVE domain played an important role in this
process.

### ZFYVE28 promoted the conversion of early endosomes to late endosomes

Following insulin binding, INSR is phosphorylated, and active INSR is
then internalized by clathrin-mediated endocytosis. Then, the receptor
endocytosed into early endosomes is either transported back to the
plasma membrane by recycling endosomes or enters the lysosomal
degradation pathway via conversion of early endosomes to late
endosomes[28,29]. We sought to explore the role of ZFYVE28 in this
process. We first performed immunofluorescence staining of HepG2
cells at different time points after insulin stimulation to test the dif-
ferent endosome markers. The results showed that ZFYVE28 over-
expression could greatly promote the generation of late endosomes
(RAB7-labeled endosomes) and inhibit the production of recycling
endosomes (RAB11-labeled endosomes); conversely, in ZFYVE28-
ΔFYVE-overexpressing cells, there were relatively few late endo-
somes, while recycling endosomes increased significantly over time
(Fig. 7b, Supplementary Fig. 8e). Then, multiplex fluorescent label
detection was performed using Opal™ 7 color kits. The results of
multiplex fluorescence staining showed that ZFYVE28 inhibited the
production of recycling endosomes, whereas late endosomes
increased and colocalized with lysosomes (Supplementary Fig. 8f).
Furthermore, we performed multiple rounds of fluorescence staining
of HepG2 cells, selected large fields of view with a large number of cells
for imaging, and quantified the results of the multiple assays. The
results further confirmed that ZFYVE28 overexpression significantly
inhibited the production of recycling endosomes, while promoting the
conversion of early endosomes to late endosomes, which colocalized
with lysosomes (Fig. 7c, d). This effect was closely related to the FYVE
domain. We next examined the protein levels of the recycling endo-
some marker RAB11 and the late endosome marker RAB7 and found
that ZFYVE28 overexpression significantly increased RAB7 levels and
decreased RAB11 levels following insulin stimulation in cells (Fig. 7e).
Combined with the results described above, ZFYVE28 colocalized with
early endosomes via the FYVE domain and promoted the conversion of
early to late endosomes.

We next verified this in mice. At different time points after
intraperitoneal injection of insulin in flox/flox mice and LKO mice, we
collected liver tissues for analysis and found that Rab7 levels were
lower, but Rab11 levels were higher in LKO mice than in flox/flox mice
(Fig. 7f). Immunohistochemical results of liver sections also showed
that Rab7 was significantly increased, while Rab11 was significantly
decreased in mice overexpressing *Zfyve28* (Fig. 7g, i, j). The result was
just the opposite in *Zfyve28* LKO mice. Compared to control mice,
*Zfyve28* LKO mice had lower Rab7 levels, but higher Rab11 levels
(Fig. 7h, k, l).

In summary, ZFYVE28 colocalized with early endosomes via the
FYVE domain and promoted the conversion of early endosomes to late
endosomes.

### ZFYVE28-mediated endosomal trafficking and degradation of phospho-INSR correlated with mouse insulin sensitivity

The above results suggested that ZFYVE28 promoted phospho-INSR
degradation by promoting the conversion of early endosomes to late
endosomes. Higher levels of the late endosome marker Rab7 observed
in the livers of *Zfyve28*-overexpressing mice and higher levels of the
recycling endosome marker Rab11 observed in the livers of *Zfyve28*-
LKO mice might account for the effect of Zfyve28 on insulin sensitivity.
However, there was a clear difference in body weight between HFD-fed
control mice and knockout mice (Figs. 4g and 5e). To exclude the
effect of body weight changes on insulin sensitivity in mice, we
enlarged the sample size and tested and compared insulin sensitivity in
8-week-old adult knockout mice and control mice fed a normal diet
(*N* = 10 per group). *Zfyve28* KO mice and LKO mice showed no apparent
differences in body weight compared to control mice (Supplementary
Fig. 9a, d). However, the GTT and ITT results showed better glucose
tolerance and insulin sensitivity in KO and LKO mice (Supplementary
Fig. 9b, c, e, f). Although the difference was not as significant as when
fed a high-fat diet, this further confirmed that *Zfyve28* knockout was
strongly associated with improved insulin sensitivity.

Phospho-INSR endocytosed into cells can be returned to the
plasma membrane by recycling endosomes[28,29]. We observed higher
levels of phospho-Insr and the recycling endosome marker Rab11 in
knockout mice (Figs. 4l, 5i and 7f, h). This suggested that *Zfyve28*
knockout promoted the production of recycling endosomes and
reduced the degradation of phospho-Insr. To further test whether this
correlates with the improved insulin sensitivity observed in knockout
mice, we employed the endosomal recycling inhibitor plinabulin to
inhibit receptor trafficking back to the plasma membrane[30]. We first
treated HepG2 cells with plinabulin for 24 h, then insulin stimulation
was administered, and plasma membrane (pMem), organelle mem-
brane (oMem), cytosol and total INSR levels were measured at differ-
ent time points after insulin stimulation. Total INSR levels did not
significantly change within 30 min of insulin stimulation; however,
plinabulin treatment significantly increased oMem INSR levels, which
were higher at both 5-min time point and 30-min time point of insulin
stimulation (Fig. 8a, b). Furthermore, plinabulin treatment resulted in a
slight decrease in pMem INSR levels at 30-min time point (Fig. 8a, b).
This suggested that plinabulin inhibited INSR recycling back to the cell
membrane. We then treated WT, KO and LKO mice fed a HFD for
10 weeks by continuous 2-week intraperitoneal injection with plina-
bulin. Plinabulin treatment did not significantly alter the body weight
of mice (Fig. 8c, f, i). GTT and ITT results showed that plinabulin
treatment significantly impaired glucose tolerance and insulin sensi-
tivity in WT mice, as well as KO and LKO mice (Fig. 8d, e, g–h, j, k).

Interestingly, plinabulin treatment resulted in significant differ-
ences in blood glucose levels of WT mice at only two time points (90-
and 120-min time points), as demonstrated by the ITT results in the WT
group (Fig. 8e). However, plinabulin treatment induced significant
differences in blood glucose levels at the 90-, 120-, and 150-min time

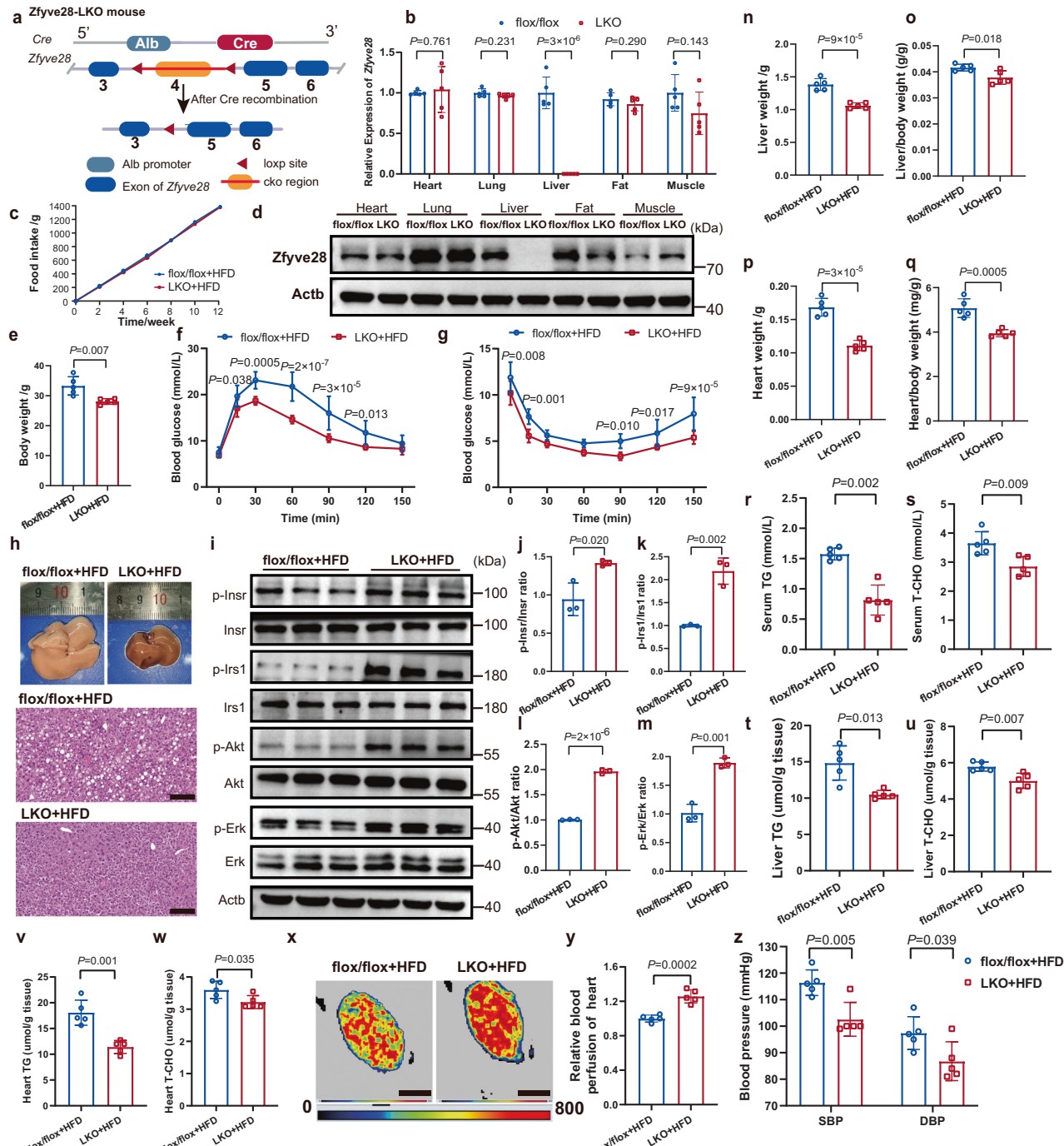

**Fig. 5 | Liver-specific *Zfyve28* knockout improved insulin sensitivity and relevant indicators associated with insulin resistance in mice. a** Schematic of the construction of liver-specific *Zfyve28* knockout mice. **b, d** qPCR (**b**, *n* = 5 biologically independent samples per group) and western blotting results (**d**, representative results of three independent parallel experiments) showed specific knockout of Zfyve28 in liver tissues. **c** Control flox/flox mice and LKO mice showed no difference in total food intake when fed a HFD; *n* = 5 per group. **e** Body weight of LKO mice and flox/flox mice; *n* = 5 biologically independent mice per group. **f, g** GTT and ITT results showed that LKO mice exhibited better glucose tolerance (**f**) and insulin sensitivity (**g**); *n* = 5 biologically independent mice per group. **h** Representative gross images and HE staining results of livers from flox/flox mice and LKO mice; *n* = 5 biologically independent mice per group. Scale bar, 100 μm. **i–m** Western blot analysis (**i**) of hepatic protein levels in flox/flox mice and LKO mice. The quantification results (**j–m**) are shown; *n* = 3 biologically independent samples per group. **n–q** Liver weight (**n**), liver/body weight ratio (**o**), heart weight (**p**), and heart/body

weight ratio (**q**) of flox/flox mice and LKO mice; *n* = 5 biologically independent samples per group. **r–w** The levels of serum TG (**r**), serum T-CHO (**s**), liver TG (**t**), liver T-CHO (**u**), heart TG (**v**) and heart T-CHO (**w**) in flox/flox mice and LKO mice; *n* = 5 biologically independent samples per group. **x-y** An increase in microcirculatory blood flow in the hearts of LKO mice was confirmed by laser Doppler blood flow measurements. Representative images (**x**) and quantification results of relative blood flow in five independent parallel experiments (**y**) are presented. Scale bar, 2 mm. **z** SBP and DBP were lower in LKO mice than in flox/flox mice; *n* = 5 biologically independent mice per group. LKO liver-specific knockout, SBP systolic blood pressure, DBP diastolic blood pressure, TG triglyceride, T-CHO total cholesterol. Data are shown as means ± SD, and *P* values are determined by unpaired two-tailed Student's *t*-test (**b**, **e**, **j–w**, **y**, **z**) or two-way ANOVA with Fisher's LSD post hoc multiple comparisons test (**f**, **g**). The exact *P* values are shown in the figure. Source data are provided as a Source Data file.

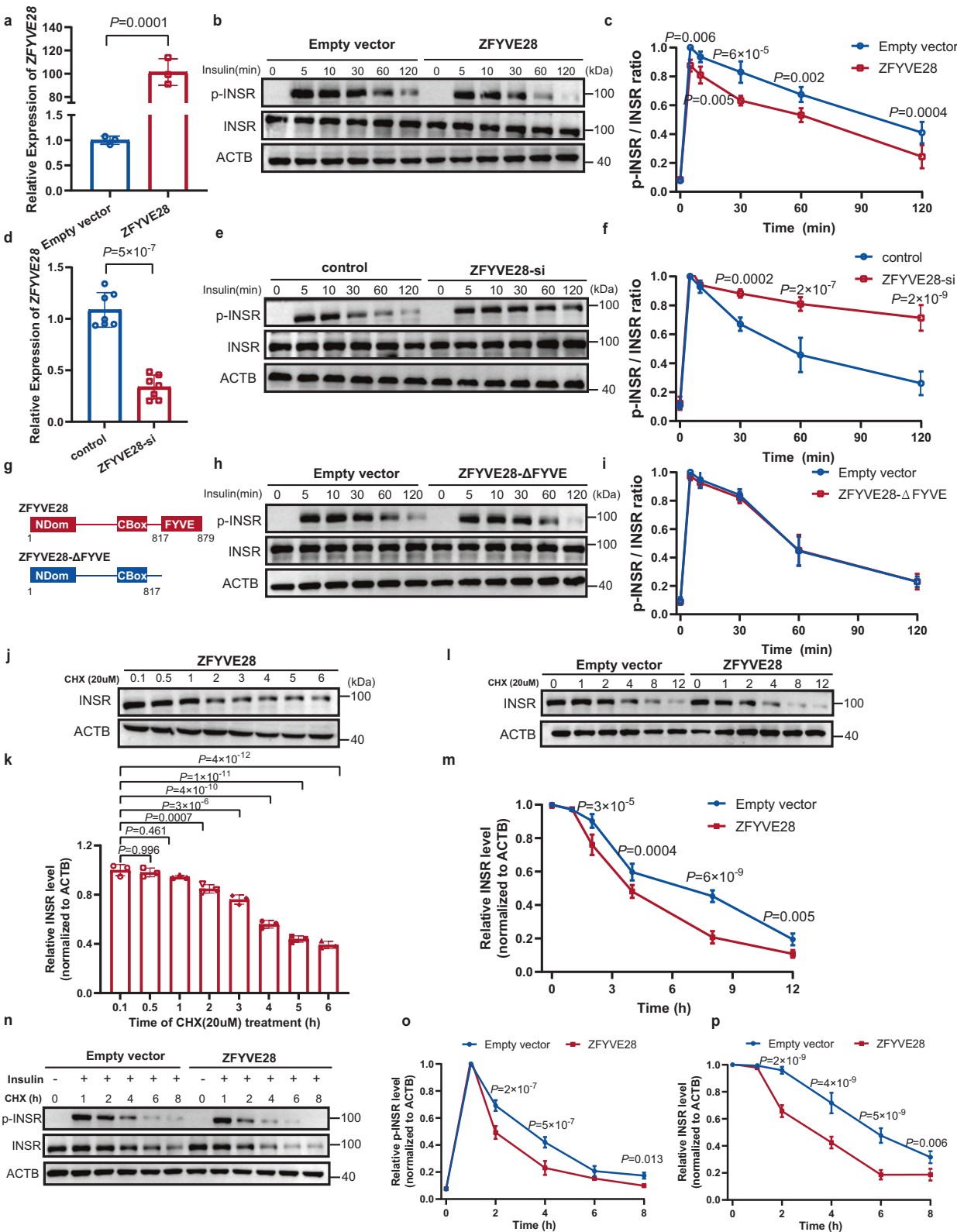

points in the ITT results of the KO group, and at the 15-, 90-, 120-, and 150-min time points in the ITT results of the LKO group (Fig. 8h, k). These results demonstrated a significant effect of plinabulin treatment on insulin sensitivity in WT, KO, and LKO mice, and the effect might be greater in KO and LKO mice than in WT mice, as indicated by the significant differences at more time points in the ITT results of KO and LKO mice.

In summary, changes in insulin sensitivity observed in KO and LKO mice were associated with Zfyve28-mediated endosomal transport and degradation of INSR but not with changes in body weight.

## Discussion

In this study, we first found lower *ZFYVE28* expression in obese patients with normal insulin sensitivity, while higher expression was observed

**Fig. 6 | ZFYVE28 promoted the degradation of phosphorylated INSR. a** ZFYVE28 was overexpressed in HepG2 cells; $n = 3$ biologically independent samples per group. **b, c** Representative western blot results of p-INSR and INSR levels (**b**) in cells overexpressing ZFYVE28; the quantitative analysis of the p-INSR/INSR ratio (**c**) at different time points after insulin stimulation at $10^{-6}$ M of three independent parallel experiments are shown. **d** ZFYVE28 was knocked down in HepG2 cells; $n = 7$ biologically independent samples per group. **e, f** The western blotting results (**e**) and the ratio of p-INSR/INSR (**f**) at different time points showed that the decrease in ZFYVE28 delayed the degradation of phosphorylated INSR. Representative western blot images and the quantitative p-INSR/INSR ratio of three independent parallel experiments are shown. **g** Schematic diagram of the construction of ZFYVE28 with the FYVE domain deleted (ZFYVE28-ΔFYVE). **h, i** Representative western blot images of p-INSR and INSR levels (**h**) in cells overexpressing ZFYVE28-ΔFYVE. The quantitative p-INSR/INSR ratio (**i**) of three independent parallel experiments are

shown. **j, k** Representative western blot images (**j**) of INSR levels in ZFYVE28-overexpressing cells treated with CHX (20 μM) for various times. The quantitative analysis results of three independent parallel experiments (**k**) are shown. **l, m** Representative western blot images of INSR levels in ZFYVE28-overexpressing cells treated with CHX (20 μM) for various times (**l**), as well as the quantitative analysis results of three independent parallel experiments (**m**). **n–p** Representative western blot images of p-INSR and INSR levels in ZFYVE28-overexpressing cells treated with insulin and CHX for various times (**n**), as well as the quantitative analysis results of three independent parallel experiments (**o, p**). Data are shown as means ± SD, and $P$ values are determined by unpaired two-tailed Student's $t$-test (**a, d**), one-way ANOVA followed by Tukey's post hoc tests (**k**) or two-way ANOVA with Fisher's LSD post hoc multiple comparisons test (**c, f, i, m, o, p**). The exact $P$ values are shown in the figure. Source data are provided as a Source Data file.

in MetS patients with insulin resistance. Moreover, Zfyve28 expression was also decreased in the liver of obese-IS mice but increased in the liver of insulin-resistant mice. We then confirmed that ZFYVE28 was transcriptionally regulated by NOTCH and inhibited by insulin through the RAS/ERK pathway. In insulin resistance, the inhibitory effect of insulin on NOTCH was attenuated due to the impairment of insulin signaling, thereby causing elevated ZFYVE28 expression. In mice, global *Zfyve28* knockout significantly improved insulin sensitivity. In addition, liver-specific *Zfyve28* overexpression in mice impaired insulin sensitivity and caused an increase in lipid content in the serum and liver, while *Zfyve28* liver-specific knockout in mice significantly improved insulin sensitivity and relevant indicators associated with insulin resistance. Mechanistically, ZFYVE28 colocalized with early endosomes via the FYVE domain. Then, ZFYVE28 inhibited the generation of recycling endosomes but promoted the conversion of early endosomes to late endosomes and targeted lysosomal degradation, which finally promoted the degradation of phosphorylated INSR. This effect disappeared with deletion of the FYVE domain (Fig. 9). We proposed the mechanism by which ZFYVE28 is involved in insulin signaling and provided a potential therapeutic target to improve insulin sensitivity and prevent metabolic and cardiovascular diseases associated with insulin resistance.

The 60% high fat diet (HFD) is one of the most commonly used diets for inducing obesity and insulin resistance in C57BL/6 male mice, but there are many variations in the induction results in different studies. He et al. investigated the effects of prolonged HFD feeding on metabolism in mice and showed that fasting body weights were not altered at weeks 4, 6 and 7 under HFD induction. Yet, 3 weeks of HFD feeding induced significant changes in GTT and ITT[31]. However, another study showed that HFD-fed C57BL/6 mice already had significant changes in body weight from week 2[32]. It was also shown that HFD-fed C57 mice had significantly increased body weight in the first week, but there was no difference in ITT at the 4-week time point[33]. Mosser et al. showed that there was no significant difference in the ITT results between ND-fed and HFD-fed mice at weeks 3, 5, 7, and 11, and 1 week of HFD feeding even improved insulin tolerance in mice[34]. The GTT results at week 5 showed meaningful small changes in blood glucose levels only at the 60-min time point, which was different from the significant large changes in blood glucose levels at multiple time points at weeks 1 and 3[34]. The homeostasis model assessment of IR (HOMA-IR) results in the 1-, 3-, 5-week HFD group were not significantly altered; and the HOMA-IR index was not always elevated after prolonged HFD induction, further confirming that the effects of HFD exposure on systemic insulin resistance are dynamically changing[31]. In addition, C57BL/6 mice fed a short-term HFD (3 days) showed sudden weight gains, whereas the rate of weight gain decreased over time[35]. Interestingly, the glucose profiles in GTT results were higher in mice fed a HFD for 5 weeks than in control mice, but slightly lower than in mice fed a HFD for only 1 week[35]. These results suggest that short-term acute HFD exposure (e.g., several days to 1 week) impairs normal

metabolism in mice, but some mice might gradually adapt over time (e.g., 4–5 weeks) and return to normal metabolic levels for a short period of time. In addition, differences between individuals in mice, which cannot be ignored, may also lead to the fact that 4–5 weeks of HFD cannot cause insulin resistance in all mice. In contrast, almost all studies have shown that chronic HFD induction for long periods (e.g., 10–16 weeks) elicits a distinct insulin-resistant phenotype in most mice[33,36–38]. In our study, we induced obesity and insulin resistance in C57BL/6 male mice by feeding a HFD for 4 and 12 weeks, respectively. Indeed, to induce an insulin-sensitive obesity phenotype in mice, we had performed parallel repeated induction experiments in multiple batches of mice. Although many of the mice fed a HFD for 4 weeks exhibited elevated body weights, impaired glucose tolerance and insulin sensitivity, a subset of mice in our study still exhibited only elevated body weights with normal glucose tolerance and insulin sensitivity. After 4 weeks of high-fat diet induction, we examined blood glucose levels in mice fasted for 4 h, and used mice whose blood glucose levels differed within 20% of the mean glucose levels in controls as candidates for insulin-sensitive obese mice. We subsequently tested insulin sensitivity of these candidate mice by GTT and ITT. In the final observation, three batches of mice (N1 = 4, N2 = 6, N3 = 3) showed normal insulin sensitivity but higher body weights than controls. We defined these mice as insulin-sensitive obese mice and chose one group ($N = 6$) as representative results to present in our data, and used them to study the expression changes of ZFYVE28 in non-insulin-resistant obese individuals, as well as the related molecular mechanisms, which was exactly the mouse model we wanted to obtain.

Insulin is a peptide hormone produced by islet β cells that acts on systemic tissues, and one of the main effects is to control systemic glucose homeostasis by regulating glucose metabolism in the liver and skeletal muscle[39–41]. Functional insulin signaling is necessary to maintain normal energy metabolism and biochemical reactions in the body[15,17]; however, disturbances in insulin signaling such as insulin resistance are partly responsible for diabetes mellitus, metabolic syndrome and cardiovascular diseases[42,43]. It has been shown that p31comet can inhibit INSR endocytosis by inhibiting the interaction of BUBR1 and INSR-bound MAD2, and liver-specific *p31comet−/−* mice exhibit reduced INSR levels and develop systemic insulin resistance, while Bubr1 deficiency delays INSR endocytosis and improves insulin sensitivity in mice[29,44]. The degradation of INSR induced by high levels of insulin stimulation has also been demonstrated to cause insulin resistance in podocytes[45,46]. The inhibition of INSR endocytosis and lysosomal degradation has been confirmed to improve insulin resistance in mice[28]. These results all suggest that endocytosis and degradation of INSR may be potential mechanisms of insulin resistance. Here, in our study, we found that ZFYVE28 expression was elevated in individuals with insulin resistance. We also demonstrated that ZFYVE28 promoted the degradation of phosphorylated INSR by promoting the conversion of early endosomes to late endosomes. In addition, accelerated insulin receptor

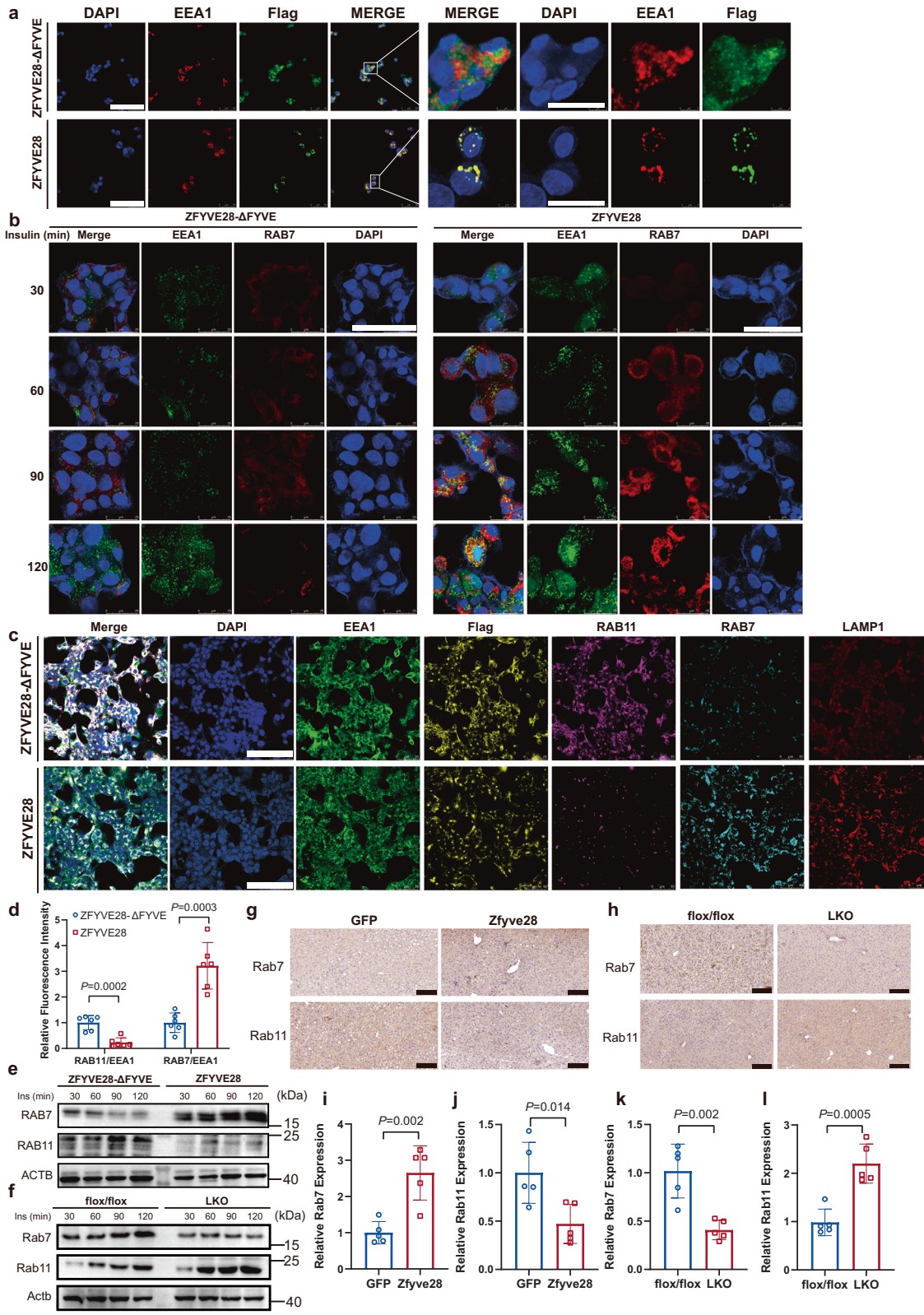

degradation has been observed in lymphocytes from patients with extreme insulin resistance[47]. This is also in line with the experimental results we obtained. In our study, while *Zfyve28* liver-specific over-expression in mice impaired insulin sensitivity and caused an increase in lipid content in the serum and liver, *Zfyve28* liver-specific knockout in mice significantly improved insulin sensitivity and other indicators associated with insulin resistance. ZFYVE28 has great potential to be a therapeutic target to slow insulin receptor degradation and improve insulin resistance.

The formation of endosomes is an important regulatory mechanism in cells that is closely related to intracellular substance transport, signal transduction, cell growth and other processes[48,49]. For receptor proteins or other substances on the plasma membrane, following plasma membrane invagination to form early endosomes,

**Fig. 7 | ZFYVE28 promoted the conversion of early endosomes to late endosomes. a** ZFYVE28 colocalized with early endosomes, whereas the colocalization disappeared with the FYVE domain deleted (ZFYVE28-ΔFYVE). Representative confocal images of four independent parallel experiments are shown, and the inset in the right panel shows a magnified view of the indicated area. Scale bar, 100 μm (left panel), 25 μm (right panel). **b** HepG2 cells overexpressing ZFYVE28 and ZFYVE28-ΔFYVE were fixed and stained at different time points after insulin stimulation at $10^{-6}$ M. ZFYVE28 promoted the formation of late endosomes (RAB7-labeled endosomes). Representative confocal images of four independent parallel experiments are shown. Scale bar, 50 μm. **c**, **d** After 90 min of insulin stimulation at $10^{-6}$ M, HepG2 cells were subjected to multiplex immunofluorescence staining using Opal™ 7 color kits (Akoya Biosciences). A large field of view was selected for confocal imaging. Representative confocal images of cells (**c**) and the quantification results of relative fluorescence density (**d**) of six independent parallel experiments are shown. Scale bar, 100 μm. **e** Representative western blot results of HepG2 cells overexpressing ZFYVE28 and ZFYVE28-ΔFYVE at different time points after insulin stimulation at $10^{-6}$ M. Independent experiments were repeated three times in parallel. **f** Representative western blot results of hepatic proteins in flox/flox mice and LKO mice after insulin injection (2 U/kg body weight) for 30, 60, 90, and 120 min. Independent experiments were repeated three times in parallel. **g**, **i**, **j** Representative results of immunohistochemical staining of liver sections from mice injected with AAV9-pTBG-GFP and AAV9-pTBG-Zfyve28 (**g**); independent experiments were repeated five times in parallel. The quantification results (**i**, **j**) are shown; $n = 5$ biologically independent samples per group. Scale bar, 100 μm. **h**, **k**, **l** Representative results of immunohistochemical staining of liver sections from flox/flox mice and LKO mice (**h**); independent experiments were repeated five times in parallel. The quantification results (**k**, **l**) are shown; $n = 5$ biologically independent samples per group. Scale bar, 100 μm. Data are shown as means ± SD, and $P$ values are determined by unpaired two-tailed Student's $t$-test (**d**, **i**–**l**). The exact $P$ values are shown in the figure. Source data are provided as a Source Data file.

some of them are subsequently retransported back to the cell surface by recycling endosomes; some are converted to late endosomes, which are then targeted to lysosomes for degradation[49–51]. This was also true during insulin signaling. Insulin acts by binding to INSR on the plasma membrane of target cells, which causes receptor tyrosine phosphorylation. In turn, INSR substrate or SHC proteins are recruited and initiate downstream cascade signaling pathways, mainly including two branches: PI3K/AKT metabolic signaling and RAS/ERK mitotic signaling[52,53]. Then the active INSR is internalized by clathrin-mediated endocytosis, and the receptor endocytosed into early endosomes is either transported back to the plasma membrane by recycling endosomes or enters the lysosomal degradation pathway via conversion of early endosomes to late endosomes[28,29]. The process of endosome formation and maturation is complex. Reticulon-3 has been shown to promote endosome maturation, and SORF-1 and SORF-2 have also been reported to play an important regulatory role in early-to-late endosome conversion[54,55]. A recent study further revealed that EPHB4 interacts with INSR and promotes the endocytosis and lysosomal degradation of INSR[28]. Here, in our study, we identified the association of ZFYVE28 with endosomal conversion. ZFYVE28 colocalized with early endosomes via its FYVE domain. Then, ZFYVE28 inhibited the generation of recycling endosomes, promoted the conversion of early endosomes to late endosomes, and targeted lysosomal degradation. This function was closely related to the FYVE domain of ZFYVE28.

The FYVE domain is a cysteine-enriched small Zn(2+) binding domain with approximately 60–70 amino acids. The structure of the FYVE domain contains two β-hairpins and a small carboxyl terminal α-helix[11,56,57]. One of the important functions of the FYVE domain is to bind PI3P and localize to endosomes. The FYVE domain contains a PI3P-binding pocket. The recognition of the 3-phosphate is determined by two clusters of conserved arginines, and the localization and activation of the PI3P-binding pocket is mediated by a nonspecific insertion of a hydrophobic loop into the lipid bilayer[11]. The interaction of the FYVE domain with PI3P is highly Zn(2+) dependent and mediates the endosomal localization and regulation of protein sorting of the FYVE domain-containing proteins[57–59]. In our study, we confirmed that the FYVE domain played an important role in the endosomal localization of ZFYVE28. With deletion of the FYVE domain, ZFYVE28 no longer colocalized with early endosomes, and the effect of promoting the conversion of early endosomes to late endosomes also disappeared.

ZFYVE28, also known as LST2 which is a major member of the FYVE protein family, has been shown to act as a negative regulator of the EGFR signaling in mammals and *C. elegans* by colocalizing with endosomes and EGF via the FYVE domain, thereby promoting the degradation of EGFR[13,27]. INSR, similar to EGFR, is an RTK. Following insulin binding, INSR is phosphorylated and activated. Then, the active INSR is internalized by clathrin-mediated endocytosis. The insulin

receptor endocytosed into early endosomes is either transported back to the plasma membrane by recycling endosomes or enters the lysosomal degradation pathway via conversion of early endosomes to late endosomes[28,29]. We found that ZFYVE28 was involved in insulin signaling and promoted the degradation of phosphorylated INSR by promoting the conversion of early endosomes to late endosomes. Moreover, some studies have proposed a strong correlation between ZFYVE28 and breast cancer, gastric cancer and colon cancer[60,61]. However, there was no evidence of obvious cancers in *Zfyve28*-over-expressing mice or knockout mice in our study. It is still an intriguing problem worth exploring that whether cancer is induced when a therapy targets ZFYVE28, which needs to be assessed in further studies.

In conclusion, we identify that ZFYVE28 is involved in insulin signaling and mediates insulin resistance by promoting phosphorylated insulin receptor degradation. While *Zfyve28* liver-specific over-expression in mice impairs insulin sensitivity and causes an increase in lipid content in the serum and liver, *Zfyve28* liver-specific knockout significantly improves insulin sensitivity and relevant indicators associated with insulin resistance. ZFYVE28 may be a potential therapeutic target to improve insulin sensitivity and prevent metabolic and cardiovascular diseases associated with insulin resistance.

## Methods

The research methods applied in this study followed the guidelines of the World Medical Association's Declaration of Helsinki and subsequent revisions. The study was reviewed and approved by the ethics committees of Fuwai Hospital and Rizhao Port Hospital.

### Cohort for expression profile chip analysis

The study population was from Rizhao City in the northern region of China from 2009 to 2010, including 100 patients with obesity, 100 patients with MetS and another 100 matched normal controls, part of which had been described previously[62–64]. The following strict inclusion criteria were used for obese patients, MetS patients and controls: (1) Chinese Han people; (2) aged 50–77 years; and (3) subjects were excluded when they had any known diseases including thyroid disease, hematological diseases, peptic ulcers, liver or kidney dysfunctions, infections, autoimmune diseases, or tumors. According to Chinese body mass index (BMI) standards[65], underweight, normal weight, and overweight/obesity were defined as BMI < 18.5, 18.5–23.9, and ≥24.0 kg/m$^2$, respectively. The controls had normal BMI (18.5–23.9), while the patients with obesity had BMI ≥ 28.0 or BMI > 27.0 and waist circumference > 101 cm. All obese patients were nondiabetic and had normal insulin sensitivity. The more strict criteria for MetS patients were BMI ≥ 28.0 or BMI > 27.0 and waist circumference > 101 cm, plus three or more of the following: (1) elevated triglycerides (drug treatment for elevated triglycerides was an alternate indicator) ≥1.7 mmol/

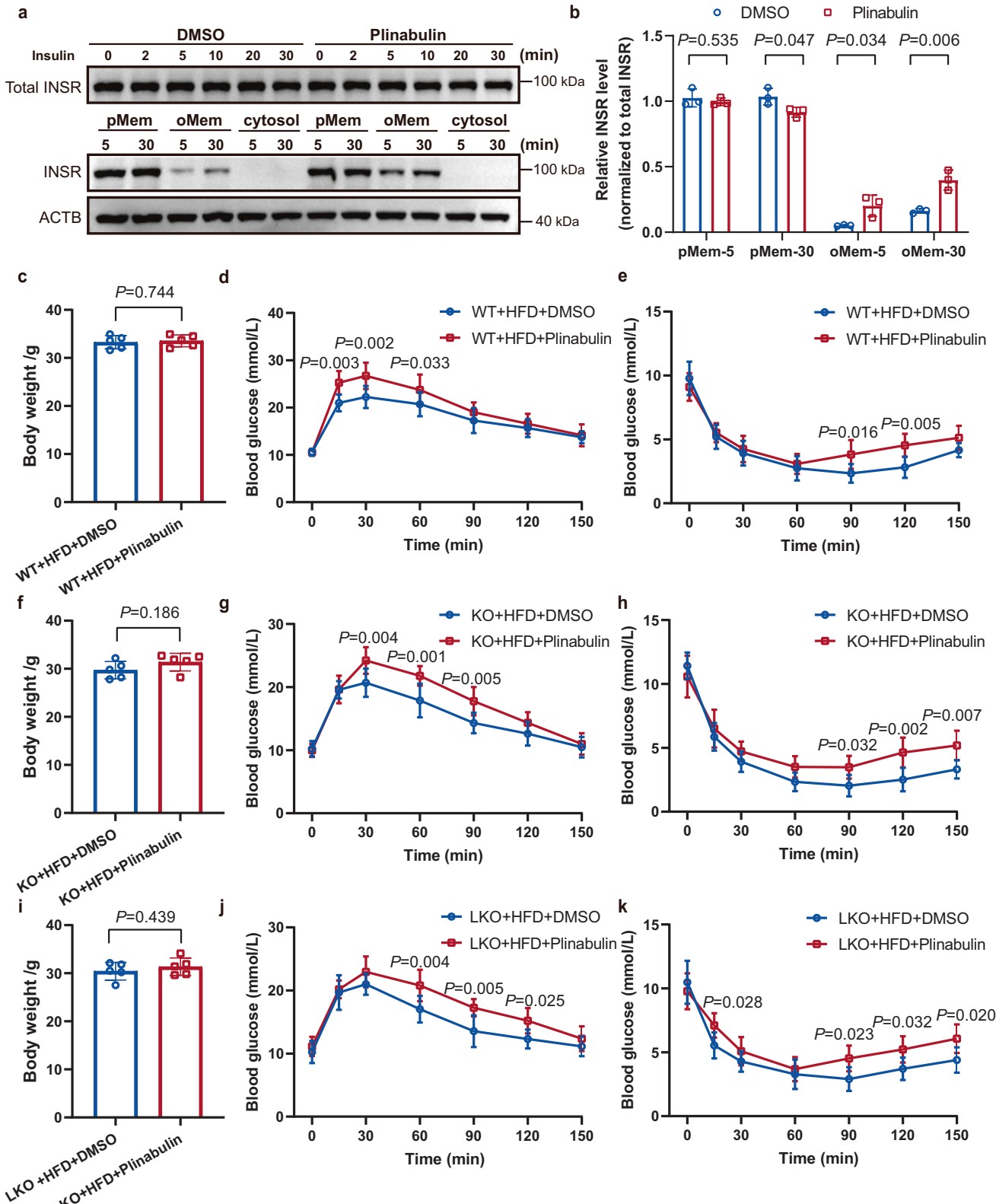

**Fig. 8 | Plinabulin treatment impaired insulin sensitivity in mice. a, b** HepG2 cells were treated with plinabulin (200 nM) for 24 h and subsequently treated with insulin (1 μM) for different times. Representative western blots of the plasma membrane (pMem), organelle membrane (oMem), cytosol and total INSR (**a**) and quantitative analysis of three independent parallel experiments (**b**) are shown. **c, f, i** Body weights of plinabulin-treated HFD-fed WT (**c**), KO (**f**), and LKO (**i**) mice; $n = 5$ biologically independent mice per group. **d, e** Plinabulin treatment impaired insulin sensitivity in HFD-fed WT mice; $n = 5$ biologically independent mice per group. **g, h, j, k** Plinabulin treatment markedly impaired insulin sensitivity in HFD-fed KO (**g, h**) and LKO mice (**j, k**); $n = 5$ biologically independent mice per group. Data are shown as means ± SD, and $P$ values are determined by unpaired two-tailed Student's $t$-test (**b, c, f, i**) or two-way ANOVA with Fisher's LSD post hoc multiple comparisons test (**d, e, g, h, j, k**). The exact $P$ values are shown in the figure. Source data are provided as a Source Data file.

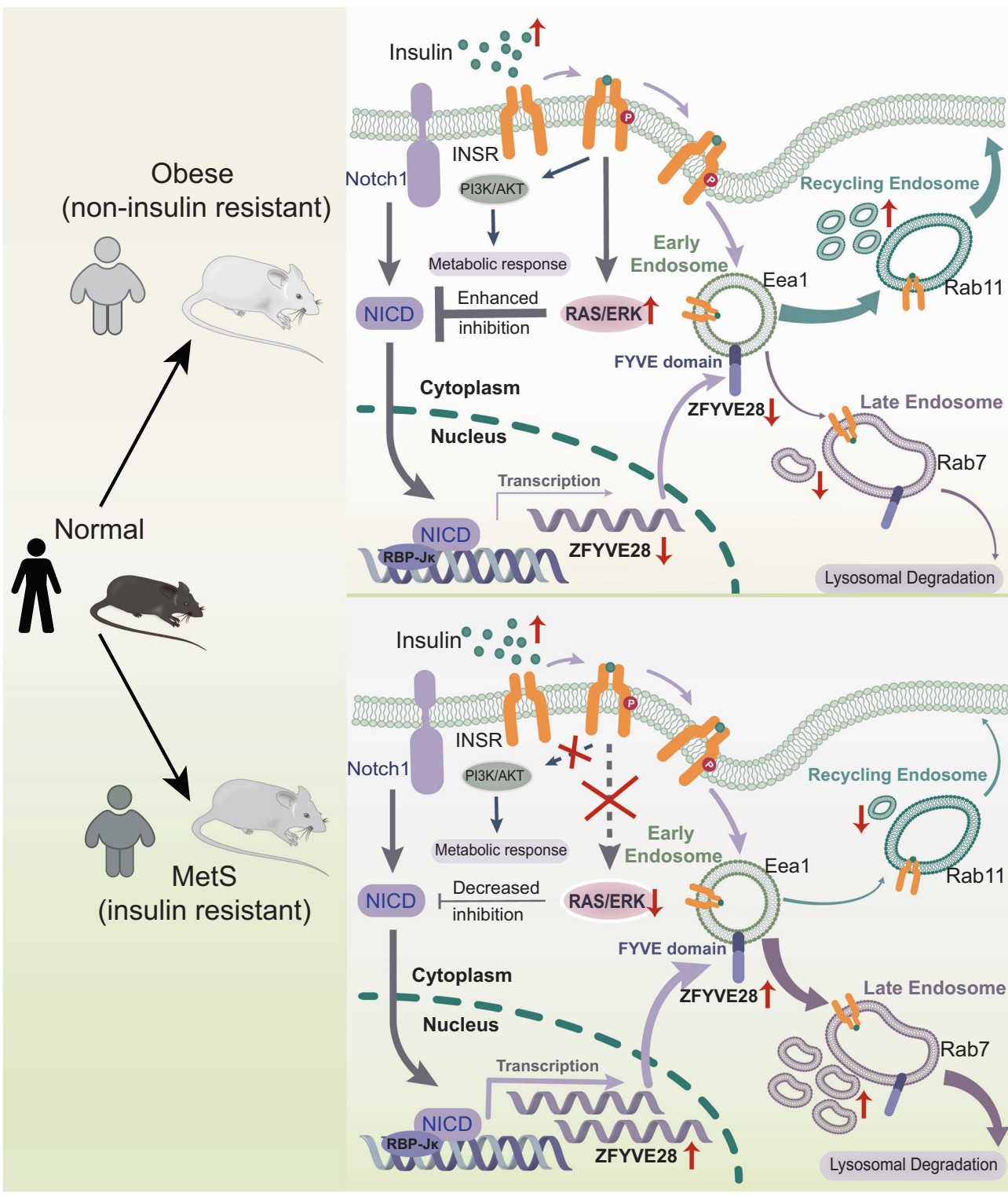

**Fig. 9 | ZFYVE28 is involved in insulin resistance and promotes phosphorylated insulin receptor degradation.** A proposed model of the mechanism by which ZFYVE28 is involved in insulin signaling. ZFYVE28 is transcriptionally regulated by NOTCH1, which is inhibited by insulin through the RAS/ERK pathway. In obese noninsulin-resistant subjects, elevated insulin levels lead to enhanced inhibition of ZFYVE28, thereby causing decreased ZFYVE28 expression. However, in insulin-resistant MetS subjects, insulin signaling is impaired and the inhibitory effect on ZFYVE28 is diminished; thus, ZFYVE28 expression is upregulated. Furthermore, ZFYVE28 colocalizes with early endosomes via its FYVE domain, inhibits the conversion of early endosomes to recycling endosomes but promotes the conversion to late endosomes, ultimately promoting the degradation of phosphorylated insulin receptor which is endocytosed into early endosomes.

L; (2) reduced HDL-C (drug treatment for reduced HDL-C was an alternate indicator) < 1.0 mmol/L; (3) elevated blood pressure (anti-hypertensive drug treatment in a patient with a history of hypertension was an alternate indicator) with current or previous SBP ≥ 160 mmHg

and DBP ≥ 100 mmHg; and (4) elevated fasting glucose (drug treatment of elevated glucose was an alternate indicator) ≥ 6.1 mmol/L. Moreover, these 100 MetS patients were all diabetic. The basic characteristics are shown in Supplementary Table 1. Then, 13 obese

patients, 13 MetS patients and 13 control males were randomly selected for gene expression profiling analysis by using Affymetrix Human Gene 2.0ST Array; the basic characteristics are shown in Supplementary Table 2. Approximately 2.5 ml of venous blood was collected with a PAXgene (Qiagen) venous blood collection tube, and blood was used to extract RNA for further research. All participants gave their signed informed consent to the study.

## Treatment of mice
All animal use and welfare adhered to the National Institutes of Health's Guide for the Care and Use of Laboratory Animals following a protocol reviewed and approved by the State Key Laboratory of Cardiovascular Disease, National Center for Cardiovascular Diseases, Fuwai Hospital (Beijing, China). The animal study was reviewed and approved by the ethics committee of Fuwai Hospital. All mice were housed under a 12-h light/dark cycle at a temperature of $23 \pm 1\,°C$ and relative humidity of 50%–60%, with free access to water. In this study, the mice used in the experiment were all male due to their lower sex hormone variations, greater susceptibility to obesity and more pronounced impairment of insulin sensitivity under high-fat diet induction[66–68]. Mice were euthanized by $CO_2$ inhalation at the appropriate time during the study and tissue samples were removed for further experiments.

For the HFD group and ND group, C57BL/6J mice were fed a high-fat diet (HFD, 60 kcal% fat, D12492, Research Diets, Inc) or normal standard chow diet with 10 kal% fat (ND, D09100304, Research Diets, Inc). Specifically, 6-week-old mice were fed a HFD for 12 weeks to induce insulin resistance; 14-week-old mice were fed a HFD for 4 weeks to induce obesity. Control mice were fed a ND continuously for 12 weeks starting at 6 weeks of age. All mice reached the experimental endpoint at 18 weeks of age.

Liver-specific overexpression of *Zfyve8* in mice was achieved by tail vein injection of AAV9-pTBG-*Zfyve28*. Control mice were injected with AAV9-pTBG-GFP. Liver-specific infection was achieved by a combination of serotype AAV9, a serotype that efficiently infects the liver, and the liver-specific promoter pTBG. Then, 6-week-old C57BL/6J mice were injected with a total volume of 100 µl PBS containing $1 \times 10^{11}$ gc virus via the tail vein. Subsequently, mice were fed a HFD for 12 weeks.

Both *Zfyve28* global knockout (KO) mice and liver-specific knockout (LKO) mice (C57BL/6J background) were constructed at Cyagen Biosciences (Suzhou, China). Liver-specific knockout of *Zfyve28* was achieved by crossing *Zfyve28*$^{flox/flox}$ mice with Alb-Cre mice. Transgenic progeny was confirmed using PCR analysis with genomic DNA isolated from the mouse tail tip. Four-week-old male mice were then fed a HFD for 12 weeks and prepared for further experiments.

For plinabulin treatment, WT, KO, and LKO mice fed a HFD for 10 weeks were used. Plinabulin (MCE, HY-14444) was diluted in DMSO and injected intraperitoneally (5 mg/kg body weight) once a day for 2 weeks. Control mice were injected with DMSO. All mice were maintained on HFD feeding at the same time, and GTT and ITT were then performed.

## Measurements of mouse blood pressure and microcirculatory blood flow of the heart
Mouse SBP and DBP were measured noninvasively using the tail cuff method (IITC Life Science, MRBP blood pressure system). Heart microcirculatory blood flow was measured with laser Doppler flowmetry (PeriCam PSI System, T402-PT).

## Glucose tolerance test (GTT) and insulin tolerance test (ITT)
Mice were fasted overnight (for GTT) or for 4 h (for ITT) before performing GTT and ITT and then intraperitoneally injected with glucose (2 g/kg body weight) for GTT or human insulin (0.75 unit/kg body weight) for ITT. Tail vein blood was taken at 0, 15, 30, 60, 90, 120 and 150 min after injection to measure blood glucose. Blood glucose levels

were measured using an Accu-Chek Active Blood Glucose Meter and corresponding test strips (Roche).

## Determination of triglycerides, total cholesterol levels and insulin levels
Triglyceride and total cholesterol contents were determined using the Triglyceride assay kit and Total cholesterol assay kit according to the manufacturer's instructions (A111-1-1, A110-1-1, Jiancheng Bioengineering Institute, Nanjing, China). Briefly, 0.1 g liver and heart tissue were weighed. Then, the tissue was homogenized thoroughly with absolute ethanol, and the supernatant was taken for detection after centrifugation. Serum insulin levels of mice in the ND group, HFD-4w group and HFD-12w group were measured using radioimmunoassay in Zhongtong Rambo (a core lab of laboratory medicine in Beijing, China).

## Isolation and culture of mouse primary hepatocytes
Isolation of mouse primary hepatocytes was performed by enzymatic digestion. Briefly, three different groups of mice fed a normal diet, a 4-week high-fat diet, and a 12-week high-fat diet were used. Primary hepatocytes were obtained by liver perfusion and digestion. Specifically, the peritoneal cavity was opened and the inferior vena cava was cannulated using a 23-gauge needle catheter. The cannula was clamped with a surgical clip. The liver was then perfused with prewarmed (37 °C) $Ca^{2+}$- and $Mg^{2+}$-free HBSS$^{++}$ containing 1 mM EGTA at a rate of 1 mL/min prior to thoracic dissection. The aorta was ligated and the portal vein was transected. The liver was then perfused at a rate of 2 mL/min with prewarmed (37 °C) HBSS$^{++}$ containing $Ca^{2+}$, $Mg^{2+}$ and collagenase digestion solution (collagen type IV, 10 mg/mL). The perfused and digested liver was then carefully dissected and placed in a petri dish containing ice-cold HBSS$^{++}$. The Glisson capsule was then gently teased with forceps to obtain the cell suspension. The cell suspension was filtered through gauze to remove undigested or connective tissue and centrifuged at $50 \times g$ for 5 min. Thereafter, the cells were resuspended in HBSS$^{++}$ and centrifuged at $50 \times g$ for 5 min to obtain parenchymal (pellet) and nonparenchymal (supernatant) cells. Parenchymal cell suspensions containing viable hepatocytes were used for culture. The hepatocytes were resuspended in prewarmed culture medium (high glucose DMEM) containing 10% fetal bovine serum (FBS). Supplements for maintenance of primary hepatocytes, including dexamethasone ITS (insulin, transferrin, selenium complex) and glutamine, were added to the medium.

## HE and immunohistochemistry staining
For HE and immunohistochemistry staining, mouse liver samples were fixed with 4% neutral buffered paraformaldehyde and paraffin embedded. Then, staining of liver sections was performed according to standard procedures.

## Cell culture
HepG2 cells (ATCC, HB-8065), HEK293T cells (ATCC, CRL-3216) and HeLa cells (ATCC, CCL-2) were cultured in DMEM (Gibco, Rockford) after processing with 10% FBS, 100 U/ml penicillin and 100 µg/ml streptomycin (P/S). Cells were used between passages 14 and 16. The cells were incubated at 37 °C (with 5% $CO_2$) until confluent and then transferred to 6-, 12- or 24-well plates. Knockdown of ZFYVE28 was mediated by siRNA transfection: sense 5′-GAC AAU GUG UGU UGA ACA UCA TT-3′; antisense 5′-UGA UGU UCA CAC AUU UGU CTT-3′. Overexpression of ZFYVE28 was mediated using lentiviral infection following standard procedures; lentiviruses with flag tags and puromycin resistance genes were constructed at GeneChem (Shanghai, China). Plasmid transfection was performed with Lipofectamine 3000 (Invitrogen, Rockford, IL) when the degree of cell fusion reached 70–80%. Alternatively, cells were treated with GSK690693 (MCE, HY-10249), Pan-RAS-IN-1 (MCE, HY-101295), KRA-533 (MCE, HY-138188) and

PD98059 (MCE, HY-12028) for the appropriate time at concentrations referred to the instructions. After incubation for the appropriate time, the cells were harvested for further experiments.

## Active Ras detection

Active Ras detection was achieved by the Active Ras Detection Kit according to the manufacturer's instructions (CST, 8821). Cell lysates were obtained under nondenaturing conditions. Optional, cell lysates were incubated with GDP or GTPγS at 30 °C for 15 min. MgCl$_2$ was then added to the mixture to terminate the reaction. The samples obtained were used as negative and positive controls during the assay. Cell lysates were subsequently affinity precipitated with GST-Raf1-RBD fusion protein. Activated Ras-GTP proteins were then precipitated and separated. Protein samples were eluted with SDS buffer containing dithiothreitol (DTT) and heated at 100 °C for 5 min. A standard western blotting assay was then performed to detect the levels of Ras-GTP.

## Extraction of plasma membrane and organelle membrane proteins

HepG2 cells were treated with plinabulin (200 nM) for 24 h and subsequently treated with insulin (1 μM) for different periods of time. Cells were washed with ice-cold PBS and then harvested. Isolation and extraction of plasma membrane and organelle membrane proteins were performed by the spin column method using the Minute™ Plasma Membrane Protein Isolation and Cell Fractionation Kit (Invent Biotechnologies, Inc., SM-005) according to the manufacturer's instructions.

## Immunofluorescence

Cells were fixed with 4% paraformaldehyde for 20–30 min at 37 °C and permeabilized with 0.2% Triton X-100 for 20 min at room temperature. After blocking with goat serum for 1 h, the cells were incubated with appropriate primary antibodies overnight at 4 °C. Alexa Fluor 488-conjugated and Alexa Fluor 594-conjugated secondary antibodies were then used. Nuclei were stained with DAPI. Confocal microscopy was performed using Lecia SP8. Before immunofluorescence staining for intracellular insulin, cells were incubated with insulin at $10^{-6}$ M. Multiplex fluorescent label detection was performed using Opal™ 7 color kits (Akoya Biosciences) following the recommended procedures. The antibodies used are shown in Table S4.

## Glucose uptake

Glucose uptake was monitored using fluorescent 2-NBDG (Invitrogen, Rockford, IL). Before staining the cells with 2-NBDG, the culture medium was removed and the cells were washed twice with prewarmed PBS. Then, freshly prepared complete high-glucose DMEM was added. Fluorescent 2-NBDG was added at a final concentration of 100 μM, and the cells were incubated at 37 °C with 5% CO$_2$ for 30 min. Then, the incubation medium was removed, and the cells were washed twice with prewarmed PBS. Fluorescence was detected using a Leica fluorescence inverted microscope (Leica DMI4000 B).

## Luciferase reporter analysis

The promoter region of the ZFYVE28 gene was cloned into the pGL3 plasmid. HepG2 cells were transfected with ZFYVE28-Luc, together with NOTCH1 and Renilla plasmids. Forty-eight hours after transfection, the luciferase activity of the cells was detected using a luciferase assay kit (Promega). The firefly luciferase and Renilla luciferase activities were determined by a microplate reader (Infinite-M200). Renilla luciferase was used for normalization.

## Chromatin immunoprecipitation (ChIP) analysis

ChIP assays were performed in HepG2 cells using the SimpleChIP® Plus Enzymatic Chromatin IP Kit (CST, 9004) following standard procedures. The prepared chromatin was reacted with anti-RBPJ (1:50, CST, 5313) and anti-IgG (1:50, CST, 2729) at 4 °C overnight. The enrichment of the DNA template was analyzed by qPCR using the following primers: forward: 5′-GAGTAGACGATGGCTGTGGG-3′ and reverse: 5′-GCCAAGTTGCTCTCACTTGC-3′, which were specific for the ZFYVE28 gene promoter.

## Quantitative RT-qPCR and western blot

To perform the RT-qPCR analysis, 1 μg of total RNA was converted to complementary DNA (cDNA). A Prism 7500 sequence-detection system (ABI, Rockford, IL) was used to analyze mRNA expression levels using Hieff® qPCR SYBR Green Master Mix. The primers are shown in Supplementary Table 3.

Proteins were extracted with IP buffer containing protease inhibitors (Beyotime, China) and the protein concentration was determined using a Pierce BCA Protein Assay Kit (Thermo Scientific). Total protein (20 μg) was subjected to SDS–PAGE, and western blotting was performed according to standard methods. The blots were then incubated with appropriate antibodies at the recommended dilutions followed by enhanced-chemiluminescent detection using the Super-Signal West Femto Maximum Sensitivity Substrate Kit (Thermo Scientific). The antibodies used are shown in Supplementary Table 4.

## Data analysis

SPSS Statistics 26.0 software was used for statistical analysis and data plotting was performed using Prism Graph Pad 8.0. The quantitative analysis of immunohistochemistry and immunofluorescence results was performed using ImageJ software, and each dot in the quantitative graphs represents a parallel independent biological sample test. For western blots, blots were converted to grayscale images, and intensity analysis was performed using ImageJ. Then, the grayscale result of the target protein (for example, ZFYVE28, NOTCH1) for each biological sample was normalized to its corresponding β-actin (ACTB) for semi-quantitative analysis, and the protein phosphorylation levels were normalized to total protein levels (for example, p-INSR/INSR, p-ERK/ERK, p-AKT/AKT). For qPCR analysis, values were the fold induction of gene expression normalized to the housekeeping gene β-actin. The differences between two groups were evaluated using unpaired Student's t tests, and multiple group comparisons were conducted by one-way ANOVA followed by Tukey's post hoc tests or two-way ANOVA with Fisher's LSD post hoc multiple comparisons test. Data are presented as the mean ± SD, and a P value < 0.05 was considered statistically significant.

## Reporting summary

Further information on research design is available in the Nature Portfolio Reporting Summary linked to this article.

## Data availability

The data supporting the findings of this study are all available in the manuscript and its supplementary information. The source data for Fig. 1b, d are provided in the Supplementary Data 1 and Supplementary Data 2 files. The RNA sequencing data for Fig. 3a–d are available at GSE245301. All other data for figures in the manuscript are provided in the Source Data file. Source data are provided with this paper.

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

## Acknowledgements

This study was supported by the Chinese Academy of Medical Sciences (Innovation Fund for Medical Sciences, 2021-I2M-1-016; Longevity Pilot Project, 2019-RC-HL-002) and National Natural Science Foundation of China (81770424 and 81970430). The authors also thank State Key Laboratory of Cardiovascular Disease, Fuwai hospital, for financial support (Opening and Operation Fund and Basic Research Fund). Y.W. received all the above grants.

## Author contributions

Y.W. designed the study and provided financial support for the study as a grantee of funds and grants. L.Y., M.X., Y.Y., S.H. Y.Z. and H.W. carried out the experiments and collected experimental data. M.Y., B.C. and X.J. collected the clinical data. M.Y. and B.C. analyzed the clinical data. L.Y., M.X., Y.Y. and S.H. conducted the animal experiments. L.Y., M.X., Y.Y., S.H. and C.L. analyzed and summarized experimental data. L.Y. wrote the paper. L.Y. and M.X. revised the manuscript. R.H. and Y.W. supervised the study. Y.W. has full access to all the data in the study. L.Y., M.X., Y.Y. and S.H. contributed equally to the study.

## Competing interests

The authors declare no competing interests.
