## [Peer Review File · Nature Communications]

ZFYVE28 mediates insulin resistance by promoting phosphorylated insulin receptor degradation via increasing late endosomes productionREVIEWER COMMENTS

Reviewer #1 (Remarks to the Author):

This study clearly demonstrated using several in vitro and in vivo (mice) experimental models the relevance of ZFYVE28 in liver insulin signaling and systemic insulin sensitivity, specifically showing that:

- ZFYVE28 was transcriptionally regulated by NOTCH and was inhibited by insulin through the RAS/ERK pathway
- While Zfyve28 global knockout and Zfyve28 liver-specific knockout in mice significantly improved insulin sensitivity and relevant insulin resistance-associated metabolic disturbances, Zfyve28 liver-specific overexpression in mice impaired insulin sensitivity and caused an increase of lipids content in the serum and liver.
- Mechanistically, ZFYVE28 co-localized with early endosomes via the FYVE domain, and inhibited the generation of recycling endosomes but promoted the conversion of early endosomes to late endosomes, targeting to lysosomal degradation, and finally resulting in the degradation of phosphorylated insulin receptor.

Authors used a large number of experiments and analyses that give a lot of consistency to the conclusions of the study, especially in liver.

My main concerns would be about the human findings and what happens to ZFYVE28 in other insulin-dependent tissues.

Major comments:

- In humans, ZFYVE28 expression was analyzed in blood. If the aim is to investigate proteins that modulate insulin sensitivity, or that explain differences between healthy obese vs insulin resistant obese, why have ZFYVE28 expression not been analyzed in insulin-dependent tissues such as adipose tissue, liver or muscle?

In humans, at least in subcutaneous adipose tissue, which of the three would be the easiest to obtain, ZFYVE28 mRNA and protein would have to be analyzed.

- In mice, ZFYVE28 mRNA and protein in adipose tissue and skeletal muscle, as well as in liver, also should be examined.

Reviewer #2 (Remarks to the Author):

Yu et al. have demonstrated a strong correlation between ZFYVE28 expression levels and insulin resistance. Using transcriptome analysis from human samples, the authors found that ZFYVE28 expression was low in insulin-sensitive obese individuals while it was high in insulin-resistant obese individuals. Using novel mouse models, including a whole body ZFYVE28 knockout mouse and a liver-specific ZFYVE28 knockout mouse, the authors suggested that ZFYVE28 might control insulin signaling in vivo. A whole body ZFYVE28 knockout mouse and a liver-specific ZFYVE28 knockout mouse are resistant to diet-induced obesity and insulin resistance. Both ZFYVE28 knockout mice demonstrated improved insulin sensitivity without altering insulin receptor (IR) levels. In contrast, liver-specific overexpression of ZFYVE28 in mice exhibited impaired insulin sensitivity.

The authors suggest that insulin signaling, specifically the RAS/MAPK pathway, inhibits NOTCH-mediated ZFYVE28 induction at transcriptional level, thus suppressing late endosome/lysosome-dependent IR degradation. In contrast, in insulin resistance conditions, insulin signaling is inactive and does not prevent NOTCH-mediated activation of ZFYVE28, which leads to IR degradation. In general, this is an interesting manuscript that uses multidisciplinary approaches including cell biology, mouse genetics, and human genomics. Large body of data demonstrate the correlation between ZFYVE28 and insulin resistance. There are, however, a number of key issues that need to be addressed:

1. In this study, the authors used the term "obese" to refer to obese individuals with "normal" insulin sensitivity. However, obesity in itself does not imply normal insulin sensitivity. Readers may be confused by this statement.
2. In this study, which sex, age, and strain of mice were used? It is important to report this information. Have the authors evaluated the phenotype of both sexes?
3. HFD-4w mice showed normal glucose and insulin tolerance (Fig. 1K-M), despite elevated serum insulin levels. This data is at odds with previous reports that indicate that one-two weeks of 60% fat HFD feeding resulted in significant insulin resistance and glucose intolerance in C57BL/6 mice. In order to explain this discrepancy, the authors should check

the genomic background of the mice used for the study and clarify this.

4. Western blot analysis should be revised and provide a clear method of quantification. The immunoblot images appear over saturated, and it is unclear how the authors analyzed the relative intensity. It is not clear what was used as a comparator for statistical significance. Please clarify this.

5. According to the method, the data are presented as a mean \pm standard deviation (SD). As an example, it is hard to believe that mice experiments show such a small variation between mice of GTT and ITT analyses (e.g. Fig. 1L, M, P, Q; 4H, I.). Moreover, functional studies in Fig. 6 and 7 are less robust. Statistics, sample size, and quantification are not provided.

6. Analysis of IR autophosphorylation and AKT phosphorylation (Fig. 2) in the same blot used to detect ZFYVE28, ERK, and Ras-GTP will further provide the information of palmitic acid (PA) on insulin sensitivity in HepG2 cells and primary hepatocytes. Can authors also examine the effect of MEK inhibitors to see the direct effect of the MAPK pathway in addition to RAS agonists or inhibitors?

Glycogen contents may be reduced due to an excessive intake of energy during PA treatment. Furthermore, glucose uptake in hepatocytes is not insulin-sensitive because they express GLUT2, not GLUT4.

7. Could you please check the ZFYVE28 levels in various mouse tissues in one blot (Fig. 4C, Fig 5D) to verify protein levels?

8. The authors performed quantitative western blot analysis on separate gels to compare insulin signaling between control and ZFYVE28 knockdown (or ZFYVE28 overexpressed cells). It is not the right method for comparing two groups. Moreover, 1 μ M insulin is not physiologically relevant.

9. It is unclear whether the decrease in pIR in Fig.6 is due to degradation or

dephosphorylation. Furthermore, there is no evidence that ZFYVE28-mediated IR degradation/dephosphorylation is responsible for the phenotype of liver-specific ZFYVE28 knockout mice. The body weight of ZFYVE28 knockout mice was significantly reduced (Figs. 4G and 5E). Perhaps this is the reason, not the consequence, for the changes in insulin sensitivity.

Reviewer #3 (Remarks to the Author):

In this manuscript, Yu et al show using a variety of assays the previously unknown role of ZFYVE28 (zinc finger FYVE-type containing 28) to regulate metabolic processes. Overall, the manuscript is very well-written, numerous mouse models generated well-described and conclusions tempered. I focus my comments primarily on the premise of the study, which needs some clarity.

1. Specifically, ZFYVE28 was identified through gene expression profiling of circulating PBMCs in patients with or without insulin resistance. How this relates to expression of this gene in liver is not evident. Is there data on expression patterns of ZFYVE28 in liver biopsies from obese patients with or without insulin resistance?

2. In wildtype, HFD-fed mice, the authors noted that ZFYVE28 expression was affected in many (but not all tissues). As the proposed mechanism for altered ZFYVE28 expression is due to changes in Notch signaling, did the authors survey these tissues for change in Notch activity relative to chow-fed control animals (analogous to Supplemental Figure 1)? Were they able to ChIP Rbp-Jk in any of these tissues, and did that change in the presence of HFD feeding?

3. Similarly, related to upstream mechanism of ZFYVE28 changes in liver, the data presented in Figure 3 are helpful to suggest that, at minimum, expression of this gene co-varies with Notch targets HES1 and HEY1. However, many of the Western blots are unclear as to what is blotted for, whether it is total Notch (and if so, which receptor - there are 4) or the active moiety, Notch-ICD. The latter would be preferred for obvious reasons, given that levels of Notch receptors may not be correlated to Notch activity. The Supplementary Table listing

antibodies used was not informative in this regard, as it is lacking catalog #s.

4. Further along these lines, while the Notch gain-of-function experiment described in Figure 3L needs clarification (what was transfected?), a loss-of-function model is necessary to establish the authors' conclusions that ZFYVE28 is transcriptionally regulated by the proposed Notch/RBP-Jk axis. There are multiple genetic and pharmacologic tools available to manipulate Notch signaling that would establish this conclusion.

Response to Reviewer 1:

1. In humans, ZFYVE28 expression was analyzed in blood. If the aim is to investigate proteins that modulate insulin sensitivity, or that explain differences between healthy obese vs insulin resistant obese, why have ZFYVE28 expression not been analyzed in insulin-dependent tissues such as adipose tissue, liver or muscle? In humans, at least in subcutaneous adipose tissue, which of the three would be the easiest to obtain, ZFYVE28 mRNA and protein would have to be analyzed.

Thank you for your helpful suggestion. We obtained 10 human fat samples (3 nonobese noninsulin-resistant samples, 4 insulin-sensitive obese samples, and 3 insulin-resistant samples) from patients who underwent liposuction surgery in Plastic Surgery Hospital, Chinese Academy of Medical Sciences, and all patients provided informed consent. We analyzed ZFYVE28 expression levels in these fat samples and found that ZFYVE28 was significantly elevated in the fat of insulin-resistant patients, which was consistent with the results observed in insulin-resistant mice (the results are shown in the figure below).

(A-C) The mRNA (A) and protein (B) expression levels of ZFYVE28 in human fat tissues (3 control samples, 4 insulin-sensitive obese samples, 3 insulin-resistant samples). The quantitative analysis results of western blots (C) are presented.

2. In mice, ZFYVE28 mRNA and protein in adipose tissue and skeletal muscle, as well as in liver, also should be examined.

Thank you for your helpful comment. We supplemented experiments to investigate Zfyve28 expression patterns in different tissues of mice and analyzed and compared changes in Zfyve28 expression levels in different tissues, including three insulin-dependent tissues, fat, liver, and skeletal muscle, in control mice, insulin-sensitive obese mice and insulin-resistant mice. The results showed that Zfyve28 was highly enriched in mouse brain tissue, which did not change significantly in obese and insulin-resistant mice. Intriguingly, Zfyve28 expression was lower in the livers of insulin-sensitive obese mice, but higher in the livers of insulin-resistant mice, both of which achieved significant differences. These are described in Figure S1 of the revised manuscript, which are also shown below.

Figure S1

(A-C) *Zfyve28* mRNA (N = 6) and protein (N = 3) expression levels in different tissues of WT mice. Representative western blots (B) and quantitative analysis results of three independent experiments (C) are shown. (D-E) Relative mRNA expression levels of *Zfyve28* in different samples of mice in the ND group (N = 3), HFD-4w group (N = 6) and HFD-12w group (N = 6). (F-G) Representative western blots of *Zfyve28* in the fat,

liver, and skeletal muscle of mice in the ND group, HFD-4w group and HFD-12w group (F). The quantitative analysis results of three independent experiments are presented (G).

We greatly appreciate your helpful and valuable comments, and we would like to thank you again for your precious time and efforts in reviewing the revised version of our manuscript.

Response to Reviewer 2:

1. In this study, the authors used the term "obese" to refer to obese individuals with "normal" insulin sensitivity. However, obesity in itself does not imply normal insulin sensitivity. Readers may be confused by this statement.

Thank you for your helpful suggestion, and we apologize for the confusion about this statement. Insulin resistance is one of the common complications in obese people and is closely related to the development of a variety of diseases, including metabolic syndrome (MetS), dyslipidemia and hypertension (1). However, although most obese patients show impaired insulin sensitivity, approximately 30% of obese patients are known to have metabolically healthy obesity in clinical observations (2). They still maintain a normal metabolic index with insulin sensitivity similar to healthy individuals, while the insulin levels are compensatorily increased (3-5). In our revised manuscript, we defined insulin-sensitive obese individuals as obese-IS individuals to avoid readers' confusion about this statement.

2. In this study, which sex, age, and strain of mice were used? It is important to report this information. Have the authors evaluated the phenotype of both sexes?

Thank you for your helpful suggestion. It has been shown that the obesity profile induced by HFD feeding is influenced by sex, age, and sex hormone levels in mice (6, 7). In our study, the mice used in the experiment were all male due to their lower sex hormone variations, greater susceptibility to obesity and more pronounced impairment of insulin sensitivity under high-fat diet induction (6, 8, 9), and unless mentioned otherwise, 6-week-old C57BL/6J mice were prepared for subsequent animal experiments. These are also described in the Methods section of the revised manuscript.

3. HFD-4w mice showed normal glucose and insulin tolerance (Fig. 1K-M), despite elevated serum insulin levels. This data is at odds with previous reports that

indicate that one-two weeks of 60% fat HFD feeding resulted in significant insulin resistance and glucose intolerance in C57BL/6 mice. In order to explain this discrepancy, the authors should check the genomic background of the mice used for the study and clarify this.

Thank you for your helpful suggestion. As reported, high-fat diet-induced obesity and insulin resistance are strongly associated with HFD exposure time. It has been shown that hepatic steatosis and changes in related metabolic gene expression are only slightly altered but not significantly altered in one- and four-week HFD-fed BALB/c mice (10). Short-term acute HFD feeding (1 week) has also been shown to induce insulin resistance in rats, as shown by significant changes in the HOMA-IR index, an indicator of insulin resistance; however, the HOMA-IR index does not differ significantly between HFD-fed mice and control mice with prolonged HFD exposure (2 or 3 weeks) (11). This suggests that insulin sensitivity may return to normal when animals are acclimated to a high-fat diet for a short period of time. In addition, C57BL/6J mice fed a short-term HFD (3 days) showed sudden weight gains, whereas the rate of weight gain decreased over time (12). A delay of 16 to 19 days in the onset of insulin resistance relative to alterations in fatty acid transport and lipid accumulation has also been reported (13). Several studies have further confirmed that chronic high-fat diet-induced behavioral and metabolic alterations in C57BL/6 mice are time-dependent (14, 15). Furthermore, it has been shown that 4-week-old young male C57BL/6J mice fed a high-fat diet for 5 weeks have significantly increased body weight, and the area under the curve (AUC) of GTT and ITT experimental results tended to be higher in HFD-fed mice than in the control, but not significantly (7). However, another study revealed that a 10-week high-fat diet induction did not significantly increase body weight in C57BL/6 mice compared with a normal diet but interfered with insulin sensitivity and glucose metabolism in C57BL/6 mice (16).

Here, in our study, we induced obesity in 6-week-old male C57BL/6J mice by feeding them a high-fat diet for 4 weeks. To maximize the possibility of excluding the effect of excessive weight differences on mouse phenotypes, we strictly controlled the body weight of mice during the experiment, and only mice with body weights within

the mean \pm 15% were included in the statistics. We examined the body weight, glucose tolerance and insulin sensitivity of these mice and the results showed that 4 weeks of high-fat diet feeding caused a significant increase in body weight, but insulin sensitivity did not change significantly. These results suggested that mice in the HFD-4w group developed an obese phenotype, but not yet to the extent of insulin resistance.

4. Western blot analysis should be revised and provide a clear method of quantification. The immunoblot images appear over saturated, and it is unclear how the authors analyzed the relative intensity. It is not clear what was used as a comparator for statistical significance. Please clarify this.

Thank you for your helpful suggestion. In our study, western blot quantification was performed with ImageJ software. Specifically, blots were converted to grayscale images and the intensity analysis was performed using ImageJ. Then, the grayscale result of the target protein (for example, ZFYVE28, NOTCH1) for each biological sample was normalized to its corresponding β -actin (ACTB) for semiquantitative analysis, and the protein phosphorylation levels were normalized to total protein levels (for example, p-INSR/INSR, p-ERK/ERK, p-AKT/AKT). For comparison of protein expression levels between the two groups, replicates of three independent biological samples were analyzed for each group and the semiquantitative results were statistically analyzed. These are also described in the Methods section of the revised manuscript.

5. According to the method, the data are presented as a mean \pm standard deviation (SD). As an example, it is hard to believe that mice experiments show such a small variation between mice of GTT and ITT analyses (e.g. Fig. 1L, M, P, Q; 4H, I.). Moreover, functional studies in Fig. 6 and 7 are less robust. Statistics, sample size, and quantification are not provided.

Thank you for your helpful suggestion. In our study, to maximize the possibility of excluding the effect of excessive weight differences on mouse phenotypes, we strictly

controlled the body weight of mice during the experiment, and only mice with body weights within the mean \pm 15% were included in the statistics. This may explain our observation of a small variation in blood glucose levels of GTT and ITT analyses in mice.

For the western blotting experiments in Figure 6, at least three independent experiments were performed with independent biological samples for each group, and representative western blots are shown in Figure 6. In addition, we performed grayscale analysis of western blot bands using ImageJ software, and quantitative analysis results of three independent experiments are presented together with representative images. For the cytofluorimetric staining results in Figure 7C, confocal images of representative fields are shown, as well as quantitative analysis results of the relative fluorescence intensity of six independent experiments analyzed using ImageJ. For immunohistochemical staining results of mouse tissues in Figure 7G and H, tissue sections from 5 independent mice were stained and examined in each group, and representative images and quantitative analysis results from 5 independent experiments are shown. These are described in Figure 6, Figure 7C-D and G-L of the revised manuscript, which are also shown below.

Figure 6

(A) ZFYVE28 was overexpressed in HepG2 cells. (B-E) Representative western blot results of p-ERK and ERK levels (B), as well as p-INSR and INSR levels (D), in cells overexpressing ZFYVE28; the quantitative analysis of the p-ERK/ERK ratio (C) and p-INSR/INSR ratio (E) at different time points after insulin stimulation at 10⁻⁶ M of three independent experiments are shown. (F) ZFYVE28 was knocked down in HepG2 cells. (G-H) The western blotting results (G) and the ratio of p-INSR/INSR (H) at different time points showed that the decrease in ZFYVE28 delayed the degradation of phosphorylated INSR. Representative western blot images and the quantitative p-INSR/INSR ratio of three independent experiments are shown. (I) Schematic diagram of the construction of ZFYVE28 with the FYVE domain deleted (ZFYVE28-ΔFYVE). (J-K) Representative western blot images of p-INSR and INSR levels (J) in cells overexpressing ZFYVE28-ΔFYVE. The quantitative p-INSR/INSR ratio (K) of three

independent experiments is shown. **(L-M)** Representative western blot images (L) of INSR levels in ZFYVE28-overexpressing cells treated with CHX (20 μ M) for various times. The quantitative analysis results of three independent experiments (M) are shown. **(N-P)** Representative western blot images of INSR levels in ZFYVE28-overexpressing cells treated with CHX (20 μ M) for various times (N, P), as well as the quantitative analysis results of three independent experiments (O, Q).

Figure 7

(C-D) After 90 minutes of insulin stimulation at 10^{-6} M, HepG2 cells were subjected to multiplex immunofluorescence staining using Opal™ 7 color kits (Akoya Biosciences). A large field of view was selected for confocal imaging. Confocal images of representative cells (C) are shown. The quantitative results (D) represent six independent tests. **(G, I-J)** Representative results of immunohistochemical staining of liver sections from mice injected with AAV9-pTBG-GFP and AAV9-pTBG-Zfyve28 (G); the quantification results (I-J) of five independent experiments are shown, and each point represents an independent biological sample (N = 5 per group). **(H, K-L)** Representative results of immunohistochemical staining of liver sections from flox/flox mice and LKO mice (G); the quantification results (K-L) of five independent experiments are shown, and each point represents an independent biological sample (N = 5 per group).

6. Analysis of IR autophosphorylation and AKT phosphorylation (Fig. 2) in the same blot used to detect ZFYVE28, ERK, and Ras-GTP will further provide the information of palmitic acid (PA) on insulin sensitivity in HepG2 cells and primary hepatocytes. Can authors also examine the effect of MEK inhibitors to see the direct effect of the MAPK pathway in addition to RAS agonists or inhibitors?

Glycogen contents may be reduced due to an excessive intake of energy during PA treatment. Furthermore, glucose uptake in hepatocytes is not insulin-sensitive because they express GLUT2, not GLUT4.

Thank you for your helpful suggestion. Palmitic acid (PA) treatment induces insulin resistance in hepatoma cells (17, 18). Although hepatocytes express higher levels of GLUT2 but not GLUT4, it has been demonstrated that insulin treatment is able to stimulate glucose uptake in hepatocytes through other pathways, which is significantly reduced in insulin-resistant hepatocytes (19-21). Therefore, we examined glucose uptake in HepG2 cells treated with PA to assess insulin resistance. Moreover, to further verify the effect of PA treatment on cellular insulin signaling, HepG2 cells were treated with PA (0.25 mM) for 48 h and then harvested to detect different protein levels by western blotting. The results showed that PA treatment significantly inhibited the phosphorylation levels of INSR, ERK and AKT, implying that insulin signaling was inhibited, accompanied by an increase in ZFYVE28 levels (Figure S3C-H). In addition, apart from the RAS agonist KRA-533 and RAS inhibitor Pan-RAS-IN-1, we also examined the direct effect of the MEK inhibitor PD98059 on ZFYVE28 and NICD, the active form of NOTCH, in HepG2 cells and primary hepatocytes from WT mice. Western blotting results showed that MEK inhibitor treatment showed similar effects as RAS inhibitor treatment, both resulting in increased levels of ZFYVE28 and NICD (Figure 2Q-S, Figure 3T-U, Figure S2O-Q, Figure S3P-Q). These are described in Figure 2Q-S, Figure 3T-U, Figure S2O-Q and Figure S3C-H, P-Q of the revised manuscript, which are also shown below.

Figure 2

(Q-S) Representative western blotting assay results (Q), as well as quantitative analysis results of three independent experiments (R-S), of primary hepatocytes from WT mice treated with insulin, KRA-533, Pan-RAS-IN-1 and PD98059.

Figure 3

(T-U) Representative western blotting results of Nicd levels, as well as quantitative analysis of three independent experiments, in primary hepatocytes from WT mice.

Figure S2

(O-Q) Representative western blot results, as well as the quantitative analysis results

of three independent experiments, of HepG2 cells treated with the MEK inhibitor PD98059.

Figure S3

(C-H) Western blot assay results of HepG2 cells treated with PA (C), as well as quantitative analysis results (D-H). (P-Q) Western blot assay results of NICD in HepG2 cells treated with Pan-RAS-IN-1, KRA-533 and PD98059 (P), as well as the quantitative analysis results (Q).

7. Could you please check the ZFYVE28 levels in various mouse tissues in one blot (Fig. 4C, Fig 5D) to verify protein levels?

Thank you for your helpful suggestion. We reperformed the western blot assays and presented the assay results on one blot. These are described in Figure 4C, Figure 5D, Figure S5B and Figure S6D of the revised manuscript, which are also shown below.

Fig. 4C**Fig. S5B****Fig. 5D****Fig. S6D**
Representative western blots of Zfyve28 expression in different tissues from *Zfyve28* knockout mice (Fig. 4C), heterozygous mice (Fig. S5B), liver-specific knockout mice (Fig. 5D), and overexpressing mice (Fig. S6D).

8. The authors performed quantitative western blot analysis on separate gels to compare insulin signaling between control and ZFYVE28 knockdown (or ZFYVE28 overexpressed cells). It is not the right method for comparing two groups. Moreover, 1 μ M insulin is not physiologically relevant.

Thank you for your helpful suggestion. In our study, we employed different concentrations of insulin to treat HepG2 cells to detect the changes in ZFYVE28 and NICD levels. The results showed that 1 μ M insulin treatment was sufficient to induce significant changes in ZFYVE28 and NICD levels (Figure S2A-D, Figure 3K). Therefore, we employed 1 μ M insulin treatment in subsequent assays. In addition, we reperformed western blotting assays and compared protein levels between two groups on one blot. These are described in Figure 6B-E, G-H and J-K of the revised manuscript, which are also shown below.

Figure 3

(K) Representative western blot results of NICD in the HepG2 cells treated with insulin; the quantitative analysis results of three independent experiments are shown.

Figure S2

(A) Relative mRNA expression of *ZFYVE28* in HepG2 cells treated with different concentrations of insulin. Data analysis was performed using one-way ANOVA followed by Tukey's post hoc tests. **(B)** Relative mRNA expression of *ZFYVE28* in HepG2, HEK293 and Hela cells treated with insulin at 10⁻⁶M. **(C-D)** Representative western blots of *ZFYVE28* in the HepG2 cells treated with different concentrations of insulin. The quantitative analysis results of three independent experiments are presented.

Figure 6

(B-E) Representative western blot results of p-ERK and ERK levels (B), as well as p-INSR and INSR levels (D), in cells overexpressing ZFYVE28; the quantitative analysis of the p-ERK/ERK ratio (C) and p-INSR/INSR ratio (E) at different time points after insulin stimulation at 10^{-6} M of three independent experiments are shown. **(G-H)** The western blotting results (G) and the ratio of p-INSR/INSR (H) at different time points showed that the decrease in ZFYVE28 delayed the degradation of phosphorylated INSR. Representative western blot images and the quantitative p-INSR/INSR ratio of three independent experiments are shown. **(J-K)** Representative western blot images of p-INSR and INSR levels (J) in cells overexpressing ZFYVE28-ΔFYVE. The quantitative p-INSR/INSR ratio (K) of three independent experiments is shown.

9. It is unclear whether the decrease in pIR in Fig.6 is due to degradation or dephosphorylation. Furthermore, there is no evidence that ZFYVE28-mediated IR degradation/dephosphorylation is responsible for the phenotype of liver-specific ZFYVE28 knockout mice. The body weight of ZFYVE28 knockout mice was significantly reduced (Figs. 4G and 5E). Perhaps this is the reason, not the consequence, for the changes in insulin sensitivity.

Thank you for your helpful suggestion. To further verify whether ZFYVE28 promoted phosphorylated INSR degradation or dephosphorylation, we treated cells overexpressing ZFYVE28 with cycloheximide (CHX) to block intracellular protein synthesis and examined total INSR levels at different time points. Western blotting results showed that INSR levels gradually decreased with increasing CHX treatment time and dropped to less than half after 6 h of CHX treatment, which implied that ZFYVE28 might promote phosphorylated INSR degradation rather than dephosphorylation (Fig. 6L-M). We then subjected ZFYVE28-overexpressing cells and control cells to prolonged CHX treatment and compared the levels of total INSR levels at different time points. The results showed that ZFYVE28 overexpression clearly promoted INSR degradation (Fig. 6N-O). Interestingly, ZFYVE28 overexpression maintained a facilitative effect on receptor degradation compared to ZFYVE28- Δ FYVE overexpression, suggesting a key role for the FYVE structural domain in this process (Fig. 6P-Q).

In addition, we supplemented relevant experiments to confirm that ZFYVE28-mediated endosomal trafficking and degradation were associated with mouse phenotypes. According to our hypothesis, ZFYVE28 colocalized with early endosomes and promoted their conversion to late endosomes, which in turn promoted phospho-INSR degradation. Higher levels of phospho-Insr and higher levels of the recycling endosome marker Rab11 observed in the livers of Zfyve28-LKO mice might account for the effect of Zfyve28 on insulin sensitivity. However, there was a clear difference in body weight between HFD-fed control mice and knockout mice. To exclude the effect of body weight changes on insulin sensitivity in mice, we enlarged the sample size and tested and compared insulin sensitivity in 8-week-old adult knockout mice and

control mice fed a normal diet (N = 10 per group). *Zfyve28* KO mice and LKO mice showed no apparent differences in body weight compared to control mice (Fig. S8A, D). However, the GTT and ITT results showed better glucose tolerance and insulin sensitivity in KO and LKO mice (Fig. S8B-C, E-F). Although the difference was not as significant as when fed a high-fat diet, this further confirmed that *Zfyve28* knockout was strongly associated with improved insulin sensitivity.

Phospho-INSR endocytosed into cells can be returned to the plasma membrane by recycling endosomes (22, 23). We observed higher levels of phospho-Insr and the recycling endosome marker Rab11 in knockout mice. This suggested that *Zfyve28* knockout promoted the production of recycling endosomes and reduced the degradation of phospho-Insr. To further test whether this correlates with the improved insulin sensitivity observed in knockout mice, we employed the endosomal recycling inhibitor plinabulin to inhibit receptor trafficking back to the plasma membrane (24). We first treated HepG2 cells with plinabulin for 24 h, then insulin stimulation was administered, and plasma membrane (pMem), organelle membrane (oMem), cytosol and total INSR levels were measured at different time points after insulin stimulation. Total INSR levels did not significantly change within 30 min of insulin stimulation; however, plinabulin treatment significantly increased oMem INSR levels, which were higher after 30 min of insulin stimulation than after 5 min of stimulation (Fig. 8A-B). Furthermore, plinabulin treatment resulted in a slight decrease in pMem INSR levels. This suggested that plinabulin inhibited INSR recycling back to the cell membrane. We then treated WT, KO and LKO mice fed a HFD for 10 weeks by continuous 2-week intraperitoneal injection with Plinabulin. Plinabulin treatment did not significantly alter the body weight of mice (Fig. 8C, F, I). GTT and ITT results showed that Plinabulin treatment only slightly impaired glucose tolerance and insulin sensitivity in WT mice (Fig. 8D-E). However, with plinabulin treatment, glucose tolerance and insulin sensitivity were markedly impaired in KO and LKO mice (Fig. 8G-H, J-K). That is, inhibition of the endosomal recycling pathway in knockout mice caused impaired insulin sensitivity. In summary, changes in insulin sensitivity observed in KO and LKO mice were associated with *Zfyve28*-mediated endosomal transport and degradation of

INSR but not with changes in body weight.

These are described in Figure 6L-Q, Figure S8 and Figure 8 of the revised manuscript, which are also shown below.

Figure 6

(L-M) Representative western blot images (L) of INSR levels in ZFYVE28-overexpressing cells treated with CHX (20 μM) for various times. The quantitative analysis results of three independent experiments (M) are shown. (N-Q) Representative western blot images of INSR levels in ZFYVE28-overexpressing cells treated with CHX (20 μM) for various times (N, P), as well as the quantitative analysis results of three independent experiments (O, Q).

Figure S8

(A, D) Body weight of 8-week-old control, KO and LKO mice fed a normal diet (ND). (B-C, E-F) GTT and ITT showed better glucose tolerance and insulin sensitivity in ND-fed KO (B-C) and LKO mice (E-F) than in control mice (n=10 per group).

Figure 8

(A-B) HepG2 cells were treated with plinabulin (200 nM) for 24 h and subsequently treated with insulin (1 μM) for different times. Representative western blots of plasma membrane, organelle membrane, cytosol and total INSR (A) and quantitative analysis of plasma membrane, organelle membrane, cytosol and total INSR (B) are shown. (C, F, I) Body weights of plinabulin-

treated HFD-fed WT (C), KO (F), and LKO (I) mice. **(D-E)** Plinabulin treatment only slightly impaired insulin sensitivity in HFD-fed WT mice (n=5 per group). **(G-H, J-K)** Plinabulin treatment markedly impaired insulin sensitivity in HFD-fed KO and LKO mice (n=5 per group).

We greatly appreciate your helpful and valuable comments, and we would like to thank you again for your precious time and efforts in reviewing the revised version of our manuscript.

Response to Reviewer 3:

1. Specifically, ZFYVE28 was identified through gene expression profiling of circulating PBMCs in patients with or without insulin resistance. How this relates to expression of this gene in liver is not evident. Is there data on expression patterns of ZFYVE28 in liver biopsies from obese patients with or without insulin resistance?

Thank you for your helpful suggestion. We obtained 10 human fat samples (3 nonobese noninsulin-resistant samples, 4 insulin-sensitive obese samples, and 3 insulin-resistant samples) from patients who underwent liposuction surgery in Plastic Surgery Hospital, Chinese Academy of Medical Sciences, and all patients provided informed consent. We analyzed ZFYVE28 expression levels in these fat samples and found that ZFYVE28 was significantly elevated in the fat of insulin-resistant patients, which was consistent with the results observed in insulin-resistant mice (the results are shown in the figure below). Unfortunately, we have not yet obtained other human samples, such as liver and skeletal muscle, due to the difficulty of obtaining clinical human samples. However, this does not affect our conclusions from in vivo and in vitro experiments.

We apologize to you for the lack of experimental data on human tissues and beg for your gentle understanding.

(A-C) The mRNA (A) and protein (B) expression levels of ZFYVE28 in human fat tissues (3 control samples, 4 insulin-sensitive obese samples, 3 insulin-resistant samples). The quantitative analysis results of western blots (C) are presented.

2. In wildtype, HFD-fed mice, the authors noted that ZFYVE28 expression was affected in many (but not all tissues). As the proposed mechanism for altered ZFYVE28 expression is due to changes in Notch signaling, did the authors survey these tissues for change in Notch activity relative to chow-fed control animals (analogous to Supplemental Figure 1)? Were they able to ChIP Rbp-Jk in any of these tissues, and did that change in the presence of HFD feeding?

Thank you for your helpful suggestion. In supplemental experiments, we examined and compared the expression levels of *Zfyve28* in three insulin-dependent tissues, including fat, liver, and skeletal muscle. The results showed that *Zfyve28* expression was significantly decreased in the livers of obese mice in the HFD-4w group, but significantly increased in the livers of insulin-resistant mice in the HFD-12w group. In addition, *Zfyve28* expression was also markedly increased in the fat tissues of insulin-resistant mice (Figure S1F-G). We also examined *Nicd* levels, the active form of Notch, in these three tissues, which were similar to the alterations in *Zfyve28* expression levels. *Nicd* levels were also significantly increased in the fat and liver of insulin-resistant mice, but decreased in the livers of obese mice. Moreover, we performed chromatin coimmunoprecipitation assays with fat, liver, and muscle tissues from mice. The results showed that binding of Rbp-Jk to the *Zfyve28* gene promoter region was decreased in obese mouse livers but elevated in insulin-resistant mouse livers compared to control mice, which was also increased in fat tissues of insulin-resistant mice. However, the binding of Rbp-Jk to the *Zfyve28* gene promoter region in muscle tissues was not significantly altered in either obese or insulin-resistant mice. These results are presented below.

(A-B) Representative western blots of Nicd in the fat, liver, and skeletal muscle of mice in the ND group, HFD-4w group and HFD-12w group (A). The quantitative analysis results of three independent experiments are presented (B). (C-D) Representative images of DNA electrophoresis results (C) from ChIP experiments, as well as quantitative analysis results of three independent experiments (D).

Figure S1

(F-G) Representative western blots of Zfyve28 in the fat, liver, and skeletal muscle of mice in the ND group, HFD-4w group and HFD-12w group (F). The quantitative analysis results of three independent experiments are presented (G).

3. Similarly, related to upstream mechanism of ZFYVE28 changes in liver, the data presented in Figure 3 are helpful to suggest that, at minimum, expression of this gene co-varies with Notch targets HES1 and HEY1. However, many of the Western blots are unclear as to what is blotted for, whether it is total Notch (and if so, which receptor - there are 4) or the active moiety, Notch-ICD. The latter would be preferred for obvious reasons, given that levels of Notch receptors may not be correlated to Notch activity. The Supplementary Table listing antibodies used was

not informative in this regard, as it is lacking catalog #s.

Thank you for your helpful suggestion, and we apologize for these confusing results. We have checked and confirmed the results of these western blots and indicated whether it was the NOTCH1 receptor or NICD active form in the revised manuscript. In addition, we supplemented relevant experiments to examine the effects of RAS inhibitor, RAS agonist and MEK inhibitor treatments on NICD in HepG2 cells and primary hepatocytes. These are presented in Figure 3K, R-W, Figure S3I-L and N-Q of the revised manuscript, which are also shown below.

Figure 3

(K) Representative western blot results of NICD in HepG2 cells treated with insulin; the quantitative analysis results of three independent experiments are shown. (R-S) Representative western blotting results of NICD levels (R) in HepG2 cells treated with insulin and KRA-533. The quantitative analysis results of three independent experiments (S) are shown. (T-W) Representative western blotting results of Nid1 levels, as well as quantitative analysis of three independent experiments, in primary hepatocytes from WT mice (T-U) and insulin-resistant mice (V-W).

Figure S3

(I-L) Representative western blots of NOTCH1 in HepG2 cells treated with Pan-RAS-IN-1 (I) and KRA-533 (J). The quantitative analysis results of three independent experiments (K-L) are shown. (N-O) Representative western blots of NOTCH1 in HepG2 cells treated with PA, insulin and KRA-533 (N), as well as the quantitative analysis results of three independent experiments (O). (P-Q) Western blot assay results of NICD in HepG2 cells treated with Pan-RAS-IN-1, KRA-533 and PD98059 (P), as well as the quantitative analysis results (Q).

4. Further along these lines, while the Notch gain-of-function experiment described in Figure 3L needs clarification (what was transfected?), a loss-of-function model is necessary to establish the authors' conclusions that ZFYVE28 is transcriptionally regulated by the proposed Notch/RBP-Jk axis. There are

multiple genetic and pharmacologic tools available to manipulate Notch signaling that would establish this conclusion.

Thank you for your helpful suggestion. In addition to overexpression of NOTCH1 plasmids, we supplemented NOTCH1 knockdown experiments. The results showed that overexpression of NOTCH1 in HepG2 cells caused an increase in the expression levels of *HES1*, *HEY1* and *ZFYVE28*, while knockdown of NOTCH1 significantly decreased the expression levels of *HES1*, *HEY1* and *ZFYVE28* (Fig. 3L-M). Furthermore, in addition to chromatin coimmunoprecipitation assays, we also constructed a dual luciferase reporter system by inserting the *ZFYVE28* gene promoter before the firefly luciferase reporter gene to verify the regulation of NOTCH1 on *ZFYVE28* transcription, and the results showed that NOTCH1 overexpression clearly enhanced luciferase activity (Fig. 3N-Q). These are presented in Figure 3L-Q of the revised manuscript, which are also shown below.

Figure 3

(L-M) The expression levels of *HES1*, *HEY1* and *ZFYVE28* following NOTCH1 overexpression (L) and NOTCH1 knockdown (M) in HepG2 cells. **(N)** Rbp-Jk binding site prediction in the *ZFYVE28* promoter region using JASPAR. **(O)** Chromatin immunoprecipitation assay showed the fold enrichment of Rbp-Jk in the *ZFYVE28* promoter region. **(P-Q)** Diagram of the dual luciferase reporter system (Q) and results

of the luciferase activity assay (P).

We greatly appreciate your helpful and valuable comments, and we would like to thank you again for your precious time and efforts in reviewing the revised version of our manuscript.

References

1. Eckel RH, Grundy SM, and Zimmet PZ. The metabolic syndrome. *Lancet (London, England)*. 2005;365(9468):1415-28.
2. Engin A. The Definition and Prevalence of Obesity and Metabolic Syndrome. *Advances in experimental medicine and biology*. 2017;960:1-17.
3. Stefan N, Häring HU, Hu FB, and Schulze MB. Metabolically healthy obesity: epidemiology, mechanisms, and clinical implications. *The lancet Diabetes & endocrinology*. 2013;1(2):152-62.
4. Magkos F. Metabolically healthy obesity: what's in a name? *The American journal of clinical nutrition*. 2019;110(3):533-9.
5. Al-Sulaiti H, Diboun I, Agha MV, Mohamed FFS, Atkin S, Dömling AS, et al. Metabolic signature of obesity-associated insulin resistance and type 2 diabetes. *Journal of translational medicine*. 2019;17(1):348.
6. Li D, Song H, Shuo L, Wang L, Xie P, Li W, et al. Gonadal white adipose tissue-derived exosomal MiR-222 promotes obesity-associated insulin resistance. *Aging*. 2020;12(22):22719-43.
7. Nishikawa S, Yasoshima A, Doi K, Nakayama H, and Uetsuka K. Involvement of sex, strain and age factors in high fat diet-induced obesity in C57BL/6J and BALB/cA mice. *Experimental animals*. 2007;56(4):263-72.
8. Qiu J, Bosch MA, Meza C, Navarro UV, Nestor CC, Wagner EJ, et al. Estradiol Protects Proopiomelanocortin Neurons Against Insulin Resistance. *Endocrinology*. 2018;159(2):647-64.
9. da Silva RP, Zampieri TT, Pedroso JA, Nagaishi VS, Ramos-Lobo AM, Furigo IC, et al. Leptin resistance is not the primary cause of weight gain associated with reduced sex hormone levels in female mice. *Endocrinology*. 2014;155(11):4226-36.
10. Nishikawa S, Sugimoto J, Okada M, Sakairi T, and Takagi S. Gene expression in livers of BALB/C and C57BL/6J mice fed a high-fat diet. *Toxicologic pathology*. 2012;40(1):71-82.
11. Abbasnejad Z, Nasserli B, Zardooz H, and Ghasemi R. Time-course study of high fat diet induced alterations in spatial memory, hippocampal JNK, P38, ERK and Akt activity. *Metabolic brain disease*. 2019;34(2):659-73.
12. Lee YS, Li P, Huh JY, Hwang IJ, Lu M, Kim JI, et al. Inflammation is necessary for long-term but not short-term high-fat diet-induced insulin resistance. *Diabetes*. 2011;60(10):2474-83.
13. Bonen A, Jain SS, Snook LA, Han XX, Yoshida Y, Buddo KH, et al. Extremely rapid increase in fatty acid transport and intramyocellular lipid accumulation but markedly delayed insulin resistance after high fat feeding in rats. *Diabetologia*. 2015;58(10):2381-91.
14. Winzell MS, and Ahrén B. The high-fat diet-fed mouse: a model for studying mechanisms and treatment of impaired glucose tolerance and type 2 diabetes. *Diabetes*. 2004;53 Suppl 3:S215-9.
15. Krishna S, Lin Z, de La Serre CB, Wagner JJ, Harn DH, Pepples LM, et al. Time-dependent behavioral, neurochemical, and metabolic dysregulation in female C57BL/6 mice caused by chronic high-fat diet intake. *Physiology & behavior*. 2016;157:196-208.
16. Li J, Wu H, Liu Y, and Yang L. High fat diet induced obesity model using four strains of mice: Kunming, C57BL/6, BALB/c and ICR. *Experimental animals*. 2020;69(3):326-35.
17. Rahman SM, Qadri I, Janssen RC, and Friedman JE. Fenofibrate and PBA prevent fatty acid-

- induced loss of adiponectin receptor and pAMPK in human hepatoma cells and in hepatitis C virus-induced steatosis. *Journal of lipid research*. 2009;50(11):2193-202.
18. Zhang G, Cai X, He L, Qin D, Li H, and Fan X. Skimmin Improves Insulin Resistance via Regulating the Metabolism of Glucose: In Vitro and In Vivo Models. *Frontiers in pharmacology*. 2020;11:540.
 19. Villalva-Pérez JM, Ramírez-Vargas MA, Serafín-Fabían JI, Ramírez M, Elena Moreno-Godínez M, Espinoza-Rojo M, et al. Characterization of Huh7 cells after the induction of insulin resistance and post-treatment with metformin. *Cytotechnology*. 2020;72(4):499-511.
 20. Watts R, Ghozlan M, Hughey CC, Johnsen VL, Shearer J, and Hittel DS. Myostatin inhibits proliferation and insulin-stimulated glucose uptake in mouse liver cells. *Biochemistry and cell biology = Biochimie et biologie cellulaire*. 2014;92(3):226-34.
 21. Yin J, Zuberi A, Gao Z, Liu D, Liu Z, and Ye J. Shilianhua extract inhibits GSK-3beta and promotes glucose metabolism. *American journal of physiology Endocrinology and metabolism*. 2009;296(6):E1275-80.
 22. Liu X, Wang K, Hou S, Jiang Q, Ma C, Zhao Q, et al. Insulin induces insulin receptor degradation in the liver through EphB4. *Nature metabolism*. 2022;4(9):1202-13.
 23. Hall C, Yu H, and Choi E. Insulin receptor endocytosis in the pathophysiology of insulin resistance. *Experimental & molecular medicine*. 2020;52(6):911-20.
 24. O'Sullivan MJ, and Lindsay AJ. The Endosomal Recycling Pathway-At the Crossroads of the Cell. *International journal of molecular sciences*. 2020;21(17).

REVIEWER COMMENTS

Reviewer #1 (Remarks to the Author):

In my view, all comments have been appropriately addressed and I only have a minor comment:

- New results about ZFYVE28 expression in human fat were not included in the revised version of the study. What is the reason?

Although the sample size of human fat is small (n= 10), I think that these new data could be mentioned in results (in first paragraph) and included in Supplemental Figure 1.

Reviewer #2 (Remarks to the Author):

1. We do not need to discuss metabolic phenotypes of other models that do not use C57BL/6(J) males under 60% high-fat diet (HFD) conditions (ref 10-16) given the well known fact that strains, sex, and diet composition are associated with insulin resistance. As an example, refs 14 and 15 used female mice. A study in ref 16 (PMID: 32188837) used a homemade HFD, which differed from the used 60% HFD for diet-induced obesity studies. In addition, in ref 16, C57BL/6 mice exhibited significant changes in blood glucose levels during GTT and ITT. In ref 7 (PMID: 17660680), ITT was not conducted (as the authors noted), but GTT demonstrated that HFD induced glucose intolerance in C57BL/6 mice.

In numerous other studies, C57BL/6 mice were shown to display increased blood glucose levels in GTT and insulin-resistant phenotypes in ITT or hyperinsulinemic clamp assays after 3 to 4 weeks of 60% HFD feeding (PMIDs: 23620060, 32885155, 33308894, 35112215, 28487366, 27760050). The authors argued, however, that 4 weeks of HFD did not lead to insulin resistance. Furthermore, they argued that despite elevated levels of serum insulin, 4 weeks of HFD did not affect fasting blood glucose levels or glucose clearance rates.

There are two aspects that require clarification. First, if 4 weeks of HFD did not lead to insulin resistance, why did hyperinsulinemia (Fig 1K) not affect glucose levels? Second, why did only the authors (to the best of my knowledge) fail to observe disrupted glucose metabolism in C57BL/6 male mice after 4 weeks of 60% HFD feeding?

2. The actin blots in Figure 2Q and Figure 3T are the same. Clarify this.

3. For Plinabulin experiments:

The authors argued that Plinabulin treatment only “slightly” impaired glucose/insulin tolerance in WT mice but it “markedly” impaired glucose/insulin tolerance in KO and LKO mice (Fig 8). How would you define "slightly" and "markedly"? As a result of the author's statistical analysis, all conditions are statistically significant.

Moreover, the representative western blot in Fig 8A did not match the quantification in Fig 8B.

4. For cycloheximide experiments:

In this revised manuscript, the authors examined IR levels in the presence of cycloheximide (CHX). They found that overexpression of ZFYVE28 promoted the reduction of IR levels in cells treated with CHX, arguing that ZFYVE28 may promote phosphorylated IR degradation rather than dephosphorylation. As far as I understand, the authors did not treat insulin in the presence of CHX. Is this the case? What is the result of pIR and total IR levels when insulin and CHX are present?

Reviewer #3 (Remarks to the Author):

The authors have answered all my points - congratulations on an interesting set of findings.

Response to Reviewer 1:

In my view, all comments have been appropriately addressed and I only have a minor comment:

- New results about ZFYVE28 expression in human fat were not included in the revised version of the study. What is the reason?

Although the sample size of human fat is small (n= 10), I think that these new data could be mentioned in results (in first paragraph) and included in Supplemental Figure 1.

Thank you for your helpful suggestion. We describe the exam results of human fat samples (n=10) in the Results section (in first paragraph), and the relevant results are presented in Supplemental Figure 1 of the revised manuscript.

We greatly appreciate your kind consideration, and we would like to thank you again for your precious time and efforts in reviewing our manuscript.

Figure S1.

(A-C) The mRNA (A) and protein (B) expression levels of ZFYVE28 in human fat tissues (3 control samples, 4 insulin-sensitive obese samples, 3 insulin-resistant samples). The quantitative analysis results of western blots (C) are presented. (D-F) Zfyve28 mRNA (N = 6) and protein (N = 3) expression levels in different tissues of WT mice. Representative western blots (E) and quantitative analysis results of three independent experiments (F) are shown. (G-H) Relative mRNA expression levels of Zfyve28 in different samples of mice in the ND group (N = 3), HFD-4w group (N = 6) and HFD-12w group (N = 6). (I-J) Representative western blots of Zfyve28 in the fat, liver, and skeletal muscle of mice in the ND group, HFD-4w group and HFD-12w group (I). The quantitative analysis results of three independent experiments are presented (J).

Response to Reviewer 2:

1. We do not need to discuss metabolic phenotypes of other models that do not use C57BL/6(J) males under 60% high-fat diet (HFD) conditions (ref 10-16) given the well known fact that strains, sex, and diet composition are associated with insulin resistance. As an example, refs 14 and 15 used female mice. A study in ref 16 (PMID: 32188837) used a homemade HFD, which differed from the used 60% HFD for diet-induced obesity studies. In addition, in ref 16, C57BL/6 mice exhibited significant changes in blood glucose levels during GTT and ITT. In ref 7 (PMID: 17660680), ITT was not conducted (as the authors noted), but GTT demonstrated that HFD induced glucose intolerance in C57BL/6 mice.

In numerous other studies, C57BL/6 mice were shown to display increased blood glucose levels in GTT and insulin-resistant phenotypes in ITT or hyperinsulinemic clamp assays after 3 to 4 weeks of 60% HFD feeding (PMIDs: 23620060, 32885155, 33308894, 35112215, 28487366, 27760050). The authors argued, however, that 4 weeks of HFD did not lead to insulin resistance. Furthermore, they argued that despite elevated levels of serum insulin, 4 weeks of HFD did not affect fasting blood glucose levels or glucose clearance rates.

There are two aspects that require clarification. First, if 4 weeks of HFD did not lead to insulin resistance, why did hyperinsulinemia (Fig 1K) not affect glucose levels? Second, why did only the authors (to the best of my knowledge) fail to observe disrupted glucose metabolism in C57BL/6 male mice after 4 weeks of 60% HFD feeding?

Thank you for your helpful suggestion, and we apologize for the puzzlement caused by the HFD-induced phenotypic changes in other strains of mice that we discussed in our previous response. We did not aim to claim that mice fed a HFD for 4 weeks did not develop insulin resistance. Our aim was simply to select obese mice that still had normal insulin sensitivity after 4 weeks of HFD induction to construct an insulin-sensitive obese mouse model. In our study, we proposed the hypothesis that high levels of insulin in physiological states inhibited the expression of ZFYVE28. In order to detect serum

insulin levels in insulin-sensitive obese mice (HFD-4w group) under physiological conditions, we did not starve the mice. Thus, Figure 1K illustrates serum insulin levels in the non-fasting state, which was higher in the HFD-4w group than that in the ND group but lower than that in the 12-week HFD-induced insulin-resistant mice shown in Figure 1O. In addition, because the obese mouse model we used in the HFD-4w group had normal insulin sensitivity and the elevated insulin levels in the non-fasting state could perform their hypoglycemic function normally, the blood glucose levels in the fasting state did not show a significant alteration after we interrupted the mice from eating, as demonstrated by the lack of a significant difference in fasting glucose levels detected at the 0-min time point in GTT and ITT of the starvation-treated mice (mice were fasted overnight before GTT and fasted for 4 h before ITT). It has also been reported that blood glucose and insulin levels are not fixed in HFD-fed C57BL/6 mice. One study revealed that fasting blood glucose levels (time point of 0 min) did not show a significant difference in GTT and ITT results in HFD-fed mice at 3, 5, and 7 weeks, and fasting insulin levels were not altered, either (1). Another study also revealed that 4- and 12-week HFD interventions did not alter fasting blood glucose levels at time point of 0 min in GTT results (2). However, there were also findings showing a significant increase in fasting blood glucose levels but no change in serum insulin levels in mice fed a HFD for 4 weeks; in contrast, in mice fed a HFD for 8 weeks, fasting blood glucose levels were unchanged but insulin levels were significantly increased (3, 4). Blood glucose and insulin levels in the non-fasting state were also measured in mice fed a HFD, and the results showed that both serum insulin levels and blood glucose levels in the non-fasting state were significantly increased from the first week in HFD-fed mice (5). These suggest that blood glucose levels and insulin levels change dynamically with HFD interventions and are influenced by fasting and non-fasting status.

Indeed, to induce an insulin-sensitive obesity phenotype in mice, we had performed parallel repeated induction experiments in multiple batches of mice. Although many of the mice fed a HFD for 4 weeks exhibited elevated body weights, impaired glucose tolerance and insulin sensitivity, a subset of mice in our study still exhibited only

elevated body weights with normal glucose tolerance and insulin sensitivity. Considering that our aim was to construct an insulin-sensitive obese mouse model, therefore, we chose these insulin-sensitive mice to study the changes in ZFYVE28 expression levels in non-insulin-resistant obese individuals, as well as the related molecular mechanisms. Actually, as reported, although a 60% high-fat diet is considered sufficient to induce obesity and insulin resistance in C57BL/6 male mice, there are many variations in the induction results in different studies. In addition, HFD-induced obesity and insulin resistance are strongly associated with HFD exposure time. The acute HFD exposure (e.g., several days to 1 week) impairs normal metabolism in mice, but some mice gradually adapt over time (e.g., 4 to 5 weeks) and return to normal metabolic levels for a short period of time. Moreover, differences between individuals in mice, which cannot be ignored, may also lead to the fact that 4 to 5 weeks of HFD cannot cause insulin resistance in all mice. We have discussed this issue in detail in the discussion section of the revised manuscript, which is also presented below.

Discussion section of the revised manuscript

The 60% high fat diet (HFD) is one of the most commonly used diets for inducing obesity and insulin resistance in C57BL/6 male mice, but there are many variations in the induction results in different studies. He et al. investigated the effects of prolonged HFD feeding on metabolism in mice and showed that fasting body weights were not altered at weeks 4, 6 and 7 under HFD induction. Yet, 3 weeks of HFD feeding induced significant changes in GTT and ITT (6). However, another study showed that HFD-fed C57BL/6 mice already had significant changes in body weight from week 2 (2). It was also shown that HFD-fed C57 mice had significantly increased body weight in the first week, but there was no difference in ITT at the 4-week time point (5). Mosser et al. showed that there was no significant difference in the ITT results between ND-fed and HFD-fed mice at weeks 3, 5, 7, and 11, and 1 week of HFD feeding even improved insulin tolerance in mice (1). The GTT results at week 5 showed meaningful small changes in blood glucose levels only at the 60-minute time point, which was different from the significant large changes in blood glucose levels at multiple time points at weeks 1 and 3 (1). The homeostasis model assessment of IR (HOMA-IR) results in the 1-, 3-, 5-week HFD group were not significantly altered; and the HOMA-IR index was not always elevated after prolonged HFD induction, further confirming that the effects of HFD exposure on systemic insulin resistance are dynamically changing (6). In addition, C57BL/6 mice fed a short-term HFD (3 days) showed sudden weight gains, whereas the rate of weight gain decreased over time (7). Interestingly, the glucose profiles in GTT results were higher in mice fed a HFD for 5 weeks than in control mice,

but slightly lower than in mice fed a HFD for only 1 week (7). These results suggest that short-term acute HFD exposure (e.g., several days to 1 week) impairs normal metabolism in mice, but some mice might gradually adapt over time (e.g., 4 to 5 weeks) and return to normal metabolic levels for a short period of time. In addition, differences between individuals in mice, which cannot be ignored, may also lead to the fact that 4 to 5 weeks of HFD cannot cause insulin resistance in all mice. In contrast, almost all studies have shown that chronic HFD induction for long periods (e.g., 10 to 16 weeks) elicits a distinct insulin-resistant phenotype in most mice (3-5, 8). In our study, we induced obesity and insulin resistance in C57BL/6 male mice by feeding a HFD for 4 and 12 weeks, respectively. Indeed, to induce an insulin-sensitive obesity phenotype in mice, we had performed parallel repeated induction experiments in multiple batches of mice. Although many of the mice fed a HFD for 4 weeks exhibited elevated body weights, impaired glucose tolerance and insulin sensitivity, a subset of mice in our study still exhibited only elevated body weights with normal glucose tolerance and insulin sensitivity. After 4 weeks of high-fat diet induction, we examined blood glucose levels in mice fasted for 4 h, and used mice whose blood glucose levels differed within 20% of the mean glucose levels in controls as candidates for insulin-sensitive obese mice. We subsequently tested insulin sensitivity of these candidate mice by GTT and ITT. In the final observation, three batches of mice (N1 = 4, N2 = 6, N3 = 3) showed normal insulin sensitivity but higher body weights than controls. We defined these mice as insulin-sensitive obese mice and chose one group (N = 6) as representative results to present in our data, and used them to study the expression changes of ZFYVE28 in non-insulin-resistant obese individuals, as well as the related molecular mechanisms, which was exactly the mouse model we wanted to obtain.

2. The actin blots in Figure 2Q and Figure 3T are the same. Clarify this.

Thank you for your helpful suggestion. Western blot results presented in Figure 2Q and Figure 3T were tested using the same set of cell samples in one experiment, only with different sample processing conditions for different lanes, as shown in Figure 2Q and Figure 3T. That is, the results presented in Figure 2Q and Figure 3T are the detection results of the same experiment, except that we show them separately in Figure 2 and Figure 3 in order to correspond with the contents of figures.

Figure 2

(Q-S) Representative western blotting assay results (Q), as well as quantitative analysis results of three independent experiments (R-S), in primary hepatocytes from WT mice treated with insulin, KRA-533, Pan-RAS-IN-1 and PD98059.

Figure 3

(T-U) Representative western blotting results of Nicd levels, as well as quantitative analysis of three independent experiments, in primary hepatocytes from WT mice.

3. For Plinabulin experiments:

The authors argued that Plinabulin treatment only “slightly” impaired glucose/insulin tolerance in WT mice but it “markedly” impaired glucose/insulin tolerance in KO and LKO mice (Fig 8). How would you define "slightly" and "markedly"? As a result of the author's statistical analysis, all conditions are statistically significant.

Moreover, the representative western blot in Fig 8A did not match the quantification in Fig 8B.

Thank you for your helpful suggestion. We apologize for the confusion caused by the results in Figure 8A and Figure 8B. The quantification in Fig 8B was the quantitative analysis result of three independent experiments, while Fig 8A showed the western blotting results of one of them. We have presented a more representative western blot in the revised manuscript. In addition, another researcher blinded to the grouping information redid the quantitative analysis of the western blotting results of three independent experiments using ImageJ software. These are shown in Figure 8A and Figure 8B of the revised manuscript, which are also shown below.

For Plinabulin experiments, we apologize for the vague expression of "slightly" and "markedly". We have changed the relevant description in the revised manuscript. GTT and ITT results showed that plinabulin treatment significantly impaired glucose tolerance and insulin sensitivity in WT mice, as well as KO and LKO mice. Interestingly, plinabulin treatment resulted in significant differences in blood glucose levels of WT mice at only two time points (15- and 30-min time points), as demonstrated by the GTT results in the WT group (Fig. 8D). However, plinabulin treatment elicited significant differences in blood glucose levels at the 30-, 60-, and 90-min time points in the GTT results of the KO group, and at the 60-, 90-, and 120-min time points in the GTT results of the LKO group (Fig. 8G, J). The ITT results were similar. Both the KO group (60-, 90-, 120-, 150-min time points) and the LKO group (15-, 90-, 120-, 150-min time points) exhibited significant differences in blood glucose levels at four time points compared to the WT group where plinabulin treatment induced significant differences in blood glucose levels at only two time points (90- and 120-min time points) (Fig. 8E, H, K). These results demonstrated a significant effect of plinabulin treatment on glucose tolerance and insulin sensitivity in WT, KO, and LKO mice, and the effect might be greater in KO and LKO mice than in WT mice, as indicated by the significant differences at more time points in the GTT and ITT results of KO and LKO mice.

These are also described in the Results section of the revised manuscript.

Figure 8.

(A-B) HepG2 cells were treated with plinabulin (200 nM) for 24 h and subsequently treated with insulin (1 μM) for different times. Representative western blots of the plasma membrane, organelle membrane, cytosol and total INSR (A) and quantitative analysis of three independent experiments (B) are shown. (C, F, I) Body weights of plinabulin-treated HFD-fed WT (C), KO (F), and LKO (I) mice. (D-E) Plinabulin treatment only slightly impaired insulin sensitivity in HFD-fed WT mice (n=5 per group). (G-H, J-K) Plinabulin treatment markedly impaired insulin sensitivity in HFD-fed KO and LKO mice (n=5 per group).

4. For cycloheximide experiments:

In this revised manuscript, the authors examined IR levels in the presence of cycloheximide (CHX). They found that overexpression of ZFYVE28 promoted the reduction of IR levels in cells treated with CHX, arguing that ZFYVE28 may promote phosphorylated IR degradation rather than dephosphorylation. As far as I understand, the authors did not treat insulin in the presence of CHX. Is this the case? What is the result of pIR and total IR levels when insulin and CHX are present?

Thank you for your helpful suggestion. In previous experiments, cells were indeed treated only with CHX and not with insulin. To further evaluate the promoting effect of ZFYVE28 on phosphorylated INSR degradation, we treated ZFYVE28-overexpressing and ZFYVE28-knockdown HepG2 cells with both insulin and CHX stimulation. The phosphorylated INSR and total INSR levels were then examined at various time points. The results showed that with prolonged CHX treatment, overexpression of ZFYVE28 significantly promoted the degradation of total INSR (Fig. 6N, P), while knockdown of ZFYVE28 significantly delayed the degradation of total INSR (Fig. S7E, G). Furthermore, as expected, overexpression of ZFYVE28 caused a decrease in phosphorylated INSR levels in response to insulin, as indicated by lower phosphorylated INSR levels at the same time points compared to controls (Fig. 6N-O). In contrast, when ZFYVE28 was knocked down, phosphorylated INSR levels at the same time point were significantly higher than that in controls (Fig. S7E-F). Overall, with CHX and insulin treatment, overexpression of ZFYVE28 promoted phosphorylated INSR degradation, as shown by lower phosphorylated INSR levels and lower total INSR levels at the same time point compared with controls, while ZFYVE28 knockdown resulted in the opposite results.

These are described in the Results section of the revised manuscript, with related results showing in Figure 6N-P and Figure S7E-G, which is also shown below.

Figure 6

(N-P) Representative western blot images of p-INSR and INSR levels in ZFYVE28-overexpressing cells treated with insulin and CHX for various times (N), as well as the quantitative analysis results of three independent experiments (O-P).

Figure S7

(E-G) Representative western blot results of p-INSR and INSR levels (E) in ZFYVE28 knockdown cells treated with insulin and CHX for various times, as well as the quantitative analysis results of three independent experiments (F-G).

We greatly appreciate your helpful and valuable comments, and we would like to thank you for your precious time and efforts in reviewing our revised manuscript again.

Response to Reviewer 3:

The authors have answered all my points - congratulations on an interesting set of findings.

Thank you for your kind consideration about our work. We greatly appreciate your precious time and efforts in reviewing our manuscript.

References

1. Mosser RE, Maulis MF, Moullé VS, Dunn JC, Carboneau BA, Arasi K, et al. High-fat diet-induced β -cell proliferation occurs prior to insulin resistance in C57BL/6J male mice. *American journal of physiology Endocrinology and metabolism*. 2015;308(7):E573-82.
2. Casimiro I, Stull ND, Tersey SA, and Mirmira RG. Phenotypic sexual dimorphism in response to dietary fat manipulation in C57BL/6J mice. *Journal of diabetes and its complications*. 2021;35(2):107795.
3. Fryklund C, Neuhaus M, Morén B, Borreguero-Muñoz A, Lundmark R, and Stenkula KG. Expansion of the Inguinal Adipose Tissue Depot Correlates With Systemic Insulin Resistance in C57BL/6J Mice. *Frontiers in cell and developmental biology*. 2022;10:942374.
4. Sharma A, Singh S, Mishra A, Rai AK, Ahmad I, Ahmad S, et al. Insulin resistance corresponds with a progressive increase in NOD1 in high fat diet-fed mice. *Endocrine*. 2022;76(2):282-93.
5. Gupta D, Jetton TL, LaRock K, Monga N, Satish B, Lausier J, et al. Temporal characterization of β cell-adaptive and -maladaptive mechanisms during chronic high-fat feeding in C57BL/6NTac mice. *The Journal of biological chemistry*. 2017;292(30):12449-59.
6. He MQ, Wang JY, Wang Y, Sui J, Zhang M, Ding X, et al. High-fat diet-induced adipose tissue expansion occurs prior to insulin resistance in C57BL/6J mice. *Chronic diseases and translational medicine*. 2020;6(3):198-207.
7. Lee YS, Li P, Huh JY, Hwang IJ, Lu M, Kim JI, et al. Inflammation is necessary for long-term but not short-term high-fat diet-induced insulin resistance. *Diabetes*. 2011;60(10):2474-83.
8. Heydemann A. An Overview of Murine High Fat Diet as a Model for Type 2 Diabetes Mellitus. *Journal of diabetes research*. 2016;2016:2902351.

REVIEWERS' COMMENTS

Reviewer #2 (Remarks to the Author):

The authors have addressed all the comments I made in my review of the original manuscript. I recommend the revised manuscript be accepted for publication.

REVIEWERS' COMMENTS

Reviewer #2 (Remarks to the Author):

The authors have addressed all the comments I made in my review of the original manuscript. I recommend the revised manuscript be accepted for publication.

Response to Reviewer 2

Thank you for your kind consideration about our work. We greatly appreciate your precious time and efforts in reviewing our manuscript.